

# Revising the stochastic iterative ensemble smoother

Patrick N. Raanes[1,2], Andreas S. Stordal[1] , and Geir Evensen[1,2]

[1]NORCE, Pb. 22 Nygårdstangen, 5838 Bergen, Norway
[2]NERSC, Thormøhlens gate 47, 5006 Bergen, Norway

**Correspondence:** para@norceresearch.no

**Abstract.** Ensemble randomized maximum likelihood (EnRML) is an iterative (stochastic) ensemble smoother, used for large and nonlinear inverse problems, such as history matching and data assimilation. Its current formulation is overly complicated and has issues with computational costs, noise, and covariance localization, even causing some practitioners to omit crucial prior information. This paper resolves these difficulties and streamlines the algorithm, without changing its output. These simplifications are achieved through the careful treatment of the linearizations and subspaces. For example, it is shown (a) how ensemble linearizations relate to average sensitivity, and (b) that the ensemble does not loose rank during updates. The paper also draws significantly on the theory of the (deterministic) iterative ensemble Kalman smoother (IEnKS). Comparative benchmarks are obtained with the Lorenz-96 model with these two smoothers and the ensemble smoother using multiple data assimilation (ES-MDA).

## 1 Introduction

Ensemble (Kalman) smoothers are approximate methods used for data assimilation (state estimation in geoscience), history matching (parameter estimation for reservoirs), and other inverse problems constrained by partial differential equations. Iterative forms of the ensemble smoother, derived from optimization perspectives, have proven useful in improving the estimation accuracy when the forward operator is nonlinear. Ensemble randomized maximum likelihood (EnRML), also known as the iterative ensemble smoother (IES), is one such method. This paper fixes several issues with EnRML, described in the following. *Readers unfamiliar with EnRML may jump to the beginning of the derivation: section 2.*

### 1.1 Ensemble randomized maximum likelihood (EnRML): obstacles

The Gauss-Newton variant of EnRML was given by Gu et al. (2007); Chen and Oliver (2012), with an important precursor being Reynolds et al. (2006). This version explicitly requires the ensemble-estimated "model sensitivity" matrix, herein denoted $\bar{\mathbf{M}}_i$. As detailed in section 3, this is problematic because $\bar{\mathbf{M}}_i$ is noisy and requires the computation of the pseudo-inverse of the "anomalies", $\mathbf{X}_i^+$, for each iteration, $i$.

A Levenberg-Marquardt variant was proposed in the landmark paper of Chen and Oliver (2013b). Its main originality is a partial resolution to the above issue by modifying the Hessian (beyond the standard trust-region step regularization): the prior ensemble covariance matrix is replaced by the posterior covariance (of iteration $i$): $\bar{\mathbf{C}}_{\boldsymbol{x}} \leftarrow \bar{\mathbf{C}}_{\boldsymbol{x},i}$. Now the Kalman gain form of




the *likelihood increment* is "vastly simplified", because the linearization $\bar{\mathbf{M}}_i$ only appears in the product $\bar{\mathbf{M}}_i \bar{\mathbf{C}}_{\boldsymbol{x},i} \bar{\mathbf{M}}_i^{\mathsf{T}}$, which
does not require $\mathbf{X}_i^+$. For the *prior increment*, on the other hand, the modification breaks its Kalman gain form. Meanwhile, the
precision matrix form, i.e. their equation 10, is already invalid because it requires the inverse of $\bar{\mathbf{C}}_{\boldsymbol{x},i}$. Still, in their equation
15, the prior increment is formulated with an inversion in ensemble space, and also unburdened of the explicit computation of
$\bar{\mathbf{M}}_i$. Intermediate explanations are lacking, but could be construed to involve approximate inversions. Another issue is that the
pseudo-inverse of $\bar{\mathbf{C}}_{\boldsymbol{x}}$ is now required (via $\mathbf{X}$), and covariance localization is further complicated.

An approximate version was therefore also proposed, where the prior mismatch term is omitted from the update formula
altogether. This is not principled, and severely aggravates the chance of over-fitting and poor prediction skill. Therefore, unless
the prior mismatch term is relatively insignificant, over-fitting must be prevented by limiting the number of steps or by clever
stopping criteria. Nevertheless, this version has received significant attention in history matching.

This paper revises EnRML; without any of the above tricks, we formulate the algorithm such that there is no explicit com-
putation of $\bar{\mathbf{M}}_i$, and show how the product $\bar{\mathbf{M}}_i \mathbf{X}$ may be computed without any pseudo-inversions of the matrix of anomalies.
Consequently, the algorithm is simplified, computationally and conceptually, and there is no longer any reason to omit the
prior increment. Moreover, the Levenberg-Marquardt variant is a trivial modification of the Gauss-Newton variant. The above
is achieved by improvements to the derivation, notably by (a) improving the understanding of the sensitivity/linearizations
involved, (b) explicitly and rigorously treating issues of rank-deficiency and subspaces, and (c) avoiding premature insertion
of singular value decompositions (SVD).

## 1.2  Iterative ensemble Kalman smoother (IEnKS)

The contributions of this paper (listed by the previous paragraph) are original, but draw heavily on the theory of the iterative
ensemble Kalman smoother (IEnKS) of Sakov et al. (2012); Bocquet and Sakov (2014). Relevant precursors include (Zupanski,
2005), as well as the iterative, extended Kalman filter (e.g., Jazwinski, 1970).

It is informally known that EnRML can be seen as a stochastic flavour of the IEnKS (Sakov et al., 2012). Indeed, while the
IEnKS update takes the form of a deterministic, "square-root" transformation, based in a single objective function, EnRML
uses stochastic, "perturbed observations", associated with an ensemble of randomized objective functions. Another notable
difference is that the IEnKS was developed in the atmospheric literature, while EnRML was developed in the literature on
subsurface flow. Thus, typically, the IEnKS is applied to (sequential) state estimation problems such as filtering, while EnRML
is applied to (batch) parameter estimation problems for physical constants and boundary conditions. As illustrated by Gu et al.
(2007), however, EnRML is easily reformulated for the sequential problem, and vice-versa for the IEnKS.

The improvements to the EnRML algorithm herein renders it very similar to the IEnKS, also in computational cost. It thus
fully establishes that EnRML is "the stochastic counterpart" to the IEnKS. In spite of the similarities, the theoretical insights
and comparative experiments of this paper should make it interesting also for readers already familiar with the IEnKS.



### 1.3 Layout

Section 2 defines the inverse problem and the idea of the randomized maximum likelihood method. Section 3 derives the new formulation of EnRML, summarized in Algorithm 1 of section 3.7. Section 4 shows benchmark experiments obtained with various iterative ensemble smoothers. Appendix A provides proofs of some of the results used in the text.

## 2   RML

Randomized maximum likelihood (RML) (Kitanidis, 1995; Oliver, 1996; Oliver et al., 2008) is an approximate solution approach to a class of inverse problems. The form of RML described here is a simplification, common for large inverse problems, which does not make use of Metropolis-Hastings techniques. This restricts the class of problems for which it is unbiased, but makes it more tractable (Oliver, 2017). A similar method was proposed and studied by Bardsley et al. (2014).

### 2.1   The inverse problem

Consider the problem of estimating an unknown, high-dimensional parameter vector $\boldsymbol{x} \in \mathbb{R}^M$, given the observation $\boldsymbol{y} \in \mathbb{R}^P$. It is assumed that

$$\boldsymbol{y} = \mathcal{M}(\boldsymbol{x}) + \boldsymbol{\delta}\,, \tag{1}$$

where the (generic) forward/observation model, $\mathcal{M}$, is known and typically nonlinear, and the observation error, $\boldsymbol{\delta}$, is random noise, giving rise to a likelihood, $p(\boldsymbol{y}|\boldsymbol{x})$.

In the Bayesian paradigm, prior information is quantified as a probability density function (pdf) called the prior, denoted
$p(\boldsymbol{x})$, and the truth, $\boldsymbol{x}$, is considered a draw thereof. The inverse problem then consists of computing and representing the posterior which, in principle, is given by pointwise multiplication:

$$p(\boldsymbol{x}|\boldsymbol{y}) \propto p(\boldsymbol{y}|\boldsymbol{x})\,p(\boldsymbol{x})\,, \tag{2}$$

quantifying the updated estimation of $\boldsymbol{x}$. Due to the noted high-dimensionality and nonlinearity, this can be challenging, necessitating approximate solutions.

The prior is assumed Gaussian, with mean $\boldsymbol{\mu_x}$ and covariance $\mathbf{C_x}$, i.e.

$$p(\boldsymbol{x}) = \mathcal{N}(\boldsymbol{x}\,|\,\boldsymbol{\mu_x}, \mathbf{C_x})$$
$$= |2\pi\mathbf{C_x}|^{-\frac{1}{2}}\,e^{-\frac{1}{2}\|\boldsymbol{x}-\boldsymbol{\mu_x}\|^2_{\mathbf{C_x}}}\,. \tag{3}$$

*For now*, the prior covariance matrix, $\mathbf{C_x}$, is assumed invertible such that the corresponding norm, $\|\boldsymbol{x}\|^2_{\mathbf{C_x}} = \boldsymbol{x}^\mathsf{T}\mathbf{C_x}^{-1}\boldsymbol{x}$, is defined. Note that vectors are taken to have column orientation, and that $\boldsymbol{x}^\mathsf{T}$ denotes the transpose.

The observation error, $\boldsymbol{\delta}$, is assumed drawn from:

$$p(\boldsymbol{\delta}) = \mathcal{N}(\boldsymbol{\delta}\,|\,\mathbf{0}, \mathbf{C_\delta})\,, \tag{4}$$





whose covariance, $\mathbf{C_\delta}$, will always be assumed invertible. Then, assuming $\boldsymbol{\delta}$ and $\boldsymbol{x}$ are independent and recalling equation (1),

$$p(\boldsymbol{y}|\boldsymbol{x}) = \mathcal{N}(\boldsymbol{y}\,|\,\mathcal{M}(\boldsymbol{x}), \mathbf{C_\delta})\,. \tag{5}$$

## 2.2 The algorithm

The Monte-Carlo approach offers a convenient representation of distributions as samples. Here, the prior is represented by the "prior ensemble", $\{\boldsymbol{x}_n\}_{n=1}^N$, whose members (sample points) are assumed independently drawn from it. RML is a relatively efficient method to approximately "condition" (i.e. implement (2) on) the prior ensemble, using optimization. Firstly, an ensemble of perturbed observations, $\{\boldsymbol{y}_n\}_{n=1}^N$, is generated as $\boldsymbol{y}_n = \boldsymbol{y} + \boldsymbol{\delta}_n$, where $\boldsymbol{\delta}_n$ is independently drawn according to equation (4).

Then, the $n$-th "randomized log-posterior", $J_{\boldsymbol{x},n}$, is defined by Bayes' rule (2), except with the prior mean and the observation replaced by the $n$-th members of the prior and observation ensembles:

$$J_{\boldsymbol{x},n}(\boldsymbol{x}) = \tfrac{1}{2}\|\boldsymbol{x} - \boldsymbol{x}_n\|_{\mathbf{C}_{\boldsymbol{x}}}^2 + \tfrac{1}{2}\|\mathcal{M}(\boldsymbol{x}) - \boldsymbol{y}_n\|_{\mathbf{C_\delta}}^2\,. \tag{6}$$

The two terms are referred to as the model mismatch (log-prior) and data mismatch (log-likelihood), respectively.

Finally, these log-posteriors are minimized. Using the Gauss-Newton iterative scheme (for example) requires (7a) its gradient

and (7b) a first-order approximation to its Hessian, both evaluated at the current iterate, labelled $\boldsymbol{x}_{n,i}$ for each member $n$ and iteration $i$. To simplify the notation, define $\boldsymbol{x}_\bullet = \boldsymbol{x}_{n,i}$. Objects evaluated at $\boldsymbol{x}_\bullet$ are similarly denoted; for instance, $\mathbf{M}_\bullet = \mathcal{M}'(\boldsymbol{x}_\bullet)$ denotes the Jacobian of $\mathcal{M}$ evaluated at $\boldsymbol{x}_\bullet$, and

$$\nabla J_\bullet = \mathbf{C}_{\boldsymbol{x}}^{-1}[\boldsymbol{x}_\bullet - \boldsymbol{x}_n] + \mathbf{M}_\bullet^\mathsf{T}\mathbf{C_\delta}^{-1}[\mathcal{M}(\boldsymbol{x}_\bullet) - \boldsymbol{y}_n]\,, \tag{7a}$$

$$\mathbf{C}_\bullet^{-1} = \mathbf{C}_{\boldsymbol{x}}^{-1} + \mathbf{M}_\bullet^\mathsf{T}\mathbf{C_\delta}^{-1}\mathbf{M}_\bullet\,. \tag{7b}$$

Application of the Gauss-Newton scheme yields:

$$\begin{aligned}
\boldsymbol{x}_{n,i+1} &= \boldsymbol{x}_\bullet - \mathbf{C}_\bullet\nabla J_\bullet \\
&= \boldsymbol{x}_\bullet + \boldsymbol{\Delta}_\bullet^{\text{prior}} + \boldsymbol{\Delta}_\bullet^{\text{lklhd}}\,,
\end{aligned} \tag{8}$$

where the prior (or model) and likelihood (or data) increments are respectively given by:

$$\boldsymbol{\Delta}_\bullet^{\text{prior}} = \mathbf{C}_\bullet\mathbf{C}_{\boldsymbol{x}}^{-1}[\boldsymbol{x}_n - \boldsymbol{x}_\bullet]\,, \tag{9a}$$

$$\boldsymbol{\Delta}_\bullet^{\text{lklhd}} = \mathbf{C}_\bullet\mathbf{M}_\bullet^\mathsf{T}\mathbf{C_\delta}^{-1}[\boldsymbol{y}_n - \mathcal{M}(\boldsymbol{x}_\bullet)]\,, \tag{9b}$$

which can be called the "precision matrix" form.

Alternatively, by corollaries of the well known Woodbury identity, the increments can be written in the "Kalman gain" form:

$$\boldsymbol{\Delta}_\bullet^{\text{prior}} = (\mathbf{I}_M - \mathbf{K}_\bullet\mathbf{M}_\bullet)[\boldsymbol{x}_n - \boldsymbol{x}_\bullet]\,, \tag{10a}$$

$$\boldsymbol{\Delta}_\bullet^{\text{lklhd}} = \mathbf{K}_\bullet[\boldsymbol{y}_n - \mathcal{M}(\boldsymbol{x}_\bullet)]\,, \tag{10b}$$



where $\mathbf{K}_\bullet$ is the gain matrix:

$$\mathbf{K}_\bullet = \mathbf{C}_{\boldsymbol{x}} \mathbf{M}_\bullet^\mathsf{T} \mathbf{C}_{\boldsymbol{y}}^{-1} \, , \tag{11}$$

with

$$\mathbf{C}_{\boldsymbol{y}} = \mathbf{M}_\bullet \mathbf{C}_{\boldsymbol{x}} \mathbf{M}_\bullet^\mathsf{T} + \mathbf{C}_{\boldsymbol{\delta}} \, . \tag{12}$$

As the subscript suggests, $\mathbf{C}_{\boldsymbol{y}}$ may be identified as the prior covariance of the observation, equation (1). Note that if $P \ll M$,

then the inversion of $\mathbf{C}_{\boldsymbol{y}}$ for the Kalman gain form is significantly cheaper than the inversion to compute $\mathbf{C}_\bullet$.

## 3   EnRML

Ensemble-RML (EnRML) is an approximation of RML where the ensemble is used in its own update, by estimating $\mathbf{C}_{\boldsymbol{x}}$ and $\mathbf{M}_\bullet$. This section derives EnRML, and gradually introduces the new improvements.

Computationally, compared to RML, EnRML offers the simultaneous benefits of working with low-rank representations

of covariances, and not requiring a tangent-linear (or adjoint) model. Both advantages will be further exploited in the new formulation of EnRML.

Concerning their sampling properties, a few points can be made. Firstly (due to the ensemble covariance), EnRML is biased for finite $N$, even for a linear-Gaussian problem, for which RML will sample the posterior correctly. This bias arises for the same reasons as in the ensemble Kalman filter (EnKF, van Leeuwen, 1999; Sacher and Bartello, 2008). Secondly (due

to the ensemble linearization), EnRML effectively smoothes the likelihood. It is therefore less prone to getting trapped in local maxima of the posterior (Chen and Oliver, 2012). Sakov et al. (2017) explain this by drawing an analogy to the secant method, as compared to the Newton method. Hence, it may reasonably be expected that EnRML yield constructive results if the probability mass of the exact posterior is concentrated around its global maximum. Although this regularity condition is rather vague, it would require that the model be "not too nonlinear" in this neighbourhood. Conversely, EnRML is wholly

inept at reflecting multimodality introduced through the likelihood, and so RML may be better suited when local modes feature prominently, as is quite common in problems of subsurface flow (Oliver and Chen, 2011). However, while RML has the ability to sample multiple modes, it is difficult to predict to what extent their relative proportions will be correct (without the costly use of Metropolis-Hastings). Further comparison of the sampling properties of RML and EnRML was done by Evensen (2018b).

### 3.1   Ensemble preliminaries

For convenience, define the concatenations:

$$\mathbf{E} = \begin{bmatrix} \boldsymbol{x}_1 \, , & \ldots & \boldsymbol{x}_n \, , & \ldots & \boldsymbol{x}_N \end{bmatrix} \in \mathbb{R}^{M \times N} \, , \tag{13}$$

$$\mathbf{D} = \begin{bmatrix} \boldsymbol{\delta}_1 \, , & \ldots & \boldsymbol{\delta}_n \, , & \ldots & \boldsymbol{\delta}_N \end{bmatrix} \in \mathbb{R}^{P \times N} \, , \tag{14}$$

which are known as the "ensemble matrix" and the "perturbation matrix", respectively.





Projections sometimes appear through the use of linear regression. We therefore recall (Trefethen and Bau, 1997) that a
(square) matrix $\mathbf{\Pi}$ is an orthogonal projector if

$$\mathbf{\Pi}\mathbf{\Pi} = \mathbf{\Pi} = \mathbf{\Pi}^\mathsf{T}. \tag{15}$$

For any matrix $\mathbf{A}$, let $\mathbf{\Pi_A}$ denote the projector whose image is the column space of $\mathbf{A}$, implying that

$$\mathbf{\Pi_A}\mathbf{A} = \mathbf{A}. \tag{16}$$

Equivalently, $\mathbf{\Pi_A^\perp}\mathbf{A} = \mathbf{0}$, where $\mathbf{\Pi_A^\perp} = \mathbf{I} - \mathbf{\Pi_A}$ is called the complementary projector. The (Moore-Penrose) pseudo-inverse,
$\mathbf{A}^+$, may be used to express the projector:

$$\mathbf{\Pi_A} = \mathbf{A}\mathbf{A}^+ = (\mathbf{A}^\mathsf{T})^+(\mathbf{A}^\mathsf{T}). \tag{17}$$

Here, the second equality follows from the first by equation (15) and $(\mathbf{A}^+)^\mathsf{T} = (\mathbf{A}^\mathsf{T})^+$. The formulae simplify further in terms
of the SVD of $\mathbf{A}$.

Now, denote $\mathbb{1} \in \mathbb{R}^N$ the (column) vector of ones, and let $\mathbf{I}_N$ be the $N$-by-$N$ identity matrix. The matrix of anomalies, $\mathbf{X}$,
is defined and computed by subtracting the ensemble mean, $\bar{\boldsymbol{x}} = \mathbf{E}\mathbb{1}/N$, from each column of $\mathbf{E}$. It should be appreciated that
this amounts to the projection:

$$\mathbf{X} = \mathbf{E} - \bar{\boldsymbol{x}}\mathbb{1}^\mathsf{T} = \mathbf{E}\mathbf{\Pi_{\mathbb{1}}^\perp}, \tag{18}$$

where $\mathbf{\Pi_{\mathbb{1}}^\perp} = \mathbf{I}_N - \mathbf{\Pi_{\mathbb{1}}}$, with $\mathbf{\Pi_{\mathbb{1}}} = \mathbb{1}\mathbb{1}^\mathsf{T}/N$.

**Definition 1** (The ensemble subspace). *The flat (i.e. affine subspace) given by:* $\{\boldsymbol{x} \in \mathbb{R}^M \ : \ [\boldsymbol{x} - \bar{\boldsymbol{x}}] \in \mathrm{col}(\mathbf{X})\}$.

Similarly to section 2, iteration index $(i > 0)$ subscripting on $\mathbf{E}$, $\mathbf{X}$, and other objects, is used to indicate that they are
conditional (i.e. posterior). The iterations are initialized with the prior ensemble: $\boldsymbol{x}_{n,0} = \boldsymbol{x}_n$.

### 3.2  The constituent estimates

The ensemble estimates of $\mathbf{C}_{\boldsymbol{x}}$ and $\mathbf{M}_{\bullet}$ are the building blocks of the EnRML algorithm. The canonical estimators are used,
namely the sample covariance (19a), and the least-squares linear regression coefficients (19b). They are denoted with the
overhead bar:

$$\bar{\mathbf{C}}_{\boldsymbol{x}} = \tfrac{1}{N-1}\mathbf{X}\mathbf{X}^\mathsf{T}, \tag{19a}$$

$$\bar{\mathbf{M}}_i = \mathcal{M}(\mathbf{E}_i)\mathbf{X}_i^+. \tag{19b}$$

The anomalies at iteration $i$ are again given by $\mathbf{X}_i = \mathbf{E}_i\mathbf{\Pi_{\mathbb{1}}^\perp}$, usually computed by subtraction of $\bar{\boldsymbol{x}}_i$. The matrix $\mathcal{M}(\mathbf{E}_i)$
is defined by the column-wise application of $\mathcal{M}$ to the ensemble members. Conventionally, $\mathcal{M}(\mathbf{E}_i)$ would also be centred
5  in equation (19b), i.e. post-multiplied by $\mathbf{\Pi_{\mathbb{1}}^\perp}$. However, this operation (and notational burden) can be neglected, because



$\Pi_{\mathbb{I}}^{\perp} \mathbf{X}_i^+ = \mathbf{X}_i^+$, which follows from $\Pi(\mathbf{A}\Pi)^+ = (\mathbf{A}\Pi)^+$ (valid for any matrix $\mathbf{A}$ and projector $\Pi$ Maciejewski and Klein, 1985).

Note that the linearization (previously $\mathbf{M}_\bullet$, now $\bar{\mathbf{M}}_i$) no longer depends on the ensemble index, $n$. Indeed, it has been called "average sensitivity" since the work of Zafari et al. (2005); Reynolds et al. (2006); Gu et al. (2007). The formula (19b) for $\bar{\mathbf{M}}_i$ is sometimes arrived at via a Taylor expansion of $\mathcal{M}$ around $\bar{\boldsymbol{x}}_i$, but this requires further, indeterminate approximations to obtain any other interpretation than $\mathcal{M}'(\bar{\boldsymbol{x}}_i)$: the Jacobian evaluated at the ensemble mean. Instead, the "average sensitivity/derivative/gradient" description suggest that

$$\bar{\mathbf{M}} \approx \frac{1}{N} \sum_{n=1}^{N} \mathcal{M}'(\boldsymbol{x}_n),\tag{20}$$

where the subscript $i$ has been temporarily dropped for clarity. However, equation (20) does not appear to have been spelled out in the literature, and the sense in which it holds has not yet been established; this is accomplished by Theorem 1.

**Theorem 1** (Regression coefficients versus derivatives). *Let $\boldsymbol{x}$ be drawn from the distribution of the ensemble (e.g., the prior or posterior of any iteration): a Gaussian. Then*

$$\lim_{N \to \infty} \bar{\mathbf{M}} = \mathbb{E}[\mathcal{M}'(\boldsymbol{x})],\tag{21}$$

*with "almost sure" convergence, and expectation ($\mathbb{E}$) in $\boldsymbol{x}$. Regularity conditions and proof in appendix A. Note: the expectation could also be defined using the ensemble itself, since $\mathbb{E}[N^{-1}\sum_n \mathcal{M}'(\boldsymbol{x}_n)] = \mathbb{E}[\mathcal{M}'(\boldsymbol{x})]$.*

Note that the generality of Theorem 1 is restricted by its Gaussianity assumption. Thus, for generality and precision, $\bar{\mathbf{M}}_i$ should simply be labelled "the least-squares (linear) fit" of $\mathcal{M}$, based on $\mathbf{E}_i$.

Finally, note that the computation (19b) of $\bar{\mathbf{M}}_i$ seemingly requires calculating a new pseudo-inverse, $\mathbf{X}_i^+$, at each iteration, $i$; this is addressed in section 3.6.

The prior covariance estimate (previously $\mathbf{C}_{\boldsymbol{x}}$, now $\bar{\mathbf{C}}_{\boldsymbol{x}}$) is *not* assumed invertible, in contrast to section 2. It is then not possible to employ the precision matrix forms (9) because $\bar{\mathbf{C}}_{\boldsymbol{x}}^{-1}$ is not defined. Using the $\bar{\mathbf{C}}_{\boldsymbol{x}}^+$ in its stead is flawed and damaging because it is zero in the directions orthogonal to the ensemble subspace, so that its use would imply that the prior is assumed infinitely uncertain (i.e. flat) as opposed to infinitely certain (like a delta function) in those directions. Instead, as shown in the following, one should employ ensemble subspace formulae, or equivalently, the Kalman gain form.

## 3.3 Estimating the Kalman gain

The ensemble estimates (19) are now substituted into the Kalman gain form of the update, equation (10) to (12). The ensemble estimate of the gain matrix, denoted $\bar{\mathbf{K}}_i$, thus becomes:

$$\begin{aligned}\bar{\mathbf{K}}_i &= \bar{\mathbf{C}}_{\boldsymbol{x}} \bar{\mathbf{M}}_i^\mathsf{T} \big(\bar{\mathbf{M}}_i \bar{\mathbf{C}}_{\boldsymbol{x}} \bar{\mathbf{M}}_i^\mathsf{T} + \mathbf{C}_{\boldsymbol{\delta}}\big)^{-1} \\ &= \mathbf{X}\mathbf{Y}_i^\mathsf{T} \big(\mathbf{Y}_i \mathbf{Y}_i^\mathsf{T} + (N{-}1)\mathbf{C}_{\boldsymbol{\delta}}\big)^{-1},\end{aligned}\tag{22}$$


where $\mathbf{Y}_i$ has been defined as the *prior* (i.e. unconditioned) anomalies, under the action of the $i$-th iterate linearization:

$$\mathbf{Y}_i = \bar{\mathbf{M}}_i \mathbf{X}. \tag{23}$$

A Woodbury corollary (again, no implicit pseudo-inverting), can be used to express $\bar{\mathbf{K}}_i$ as:

$$\bar{\mathbf{K}}_i = \mathbf{X}\bar{\mathbf{C}}_{\boldsymbol{w},i}\mathbf{Y}_i^{\mathsf{T}}\mathbf{C}_{\boldsymbol{\delta}}^{-1}, \tag{24}$$

with

$$\bar{\mathbf{C}}_{\boldsymbol{w},i} = \left(\mathbf{Y}_i^{\mathsf{T}}\mathbf{C}_{\boldsymbol{\delta}}^{-1}\mathbf{Y}_i + (N-1)\mathbf{I}_N\right)^{-1}. \tag{25}$$

The reason for labelling this matrix with the subscript $\boldsymbol{w}$ is revealed later. For now, note that, in the common case of $N \ll P$, the inversion in equation (25) is significantly cheaper than the inversion in equation (22). Another computational benefit is that $\bar{\mathbf{C}}_{\boldsymbol{w},i}$ is non-dimensional, meaning that data with small magnitude will not be "perceived" as noise by numerical decomposition routines.

In conclusion, the likelihood increment (10b) is now estimated as:

$$\bar{\boldsymbol{\Delta}}_{\bullet}^{\mathrm{lklhd}} = \bar{\mathbf{K}}_i[\boldsymbol{y}_n - \mathcal{M}(\boldsymbol{x}_{\bullet})]. \tag{26}$$

This is efficient because $\bar{\mathbf{M}}_i$ does not explicitly appear in $\bar{\mathbf{K}}_i$ (neither in formula (22) nor (24)), even though it is implicitly present through $\mathbf{Y}_i$ (23), where it multiplies $\mathbf{X}$. This absence (a) is reassuring, as the product $\mathbf{Y}_i$ constitutes a less noisy estimate than just $\bar{\mathbf{M}}_i$ alone (Chen and Oliver, 2012; Emerick and Reynolds, 2013b, figures 2 and 27, resp.); (b) constitutes a

computational advantage, as will be shown in section 3.6; (c) enables leaving the type of linearization made for $\mathcal{M}$ unspecified, as is usually the case in EnKF literature.

### 3.4 Estimating the prior increment

In contrast to the likelihood increment (10b), the Kalman gain form of the prior increment (10a) explicitly contains the sensitivity matrix, $\mathbf{M}_{\bullet}$. In response, consider the change of variables:

$$\boldsymbol{x}(\boldsymbol{w}) = \bar{\boldsymbol{x}} + \mathbf{X}\boldsymbol{w}, \tag{27}$$

where $\boldsymbol{w} \in \mathbb{R}^N$ is called the ensemble "controls" (Bannister, 2016), also known as the ensemble "weights" (Ott et al., 2004), or "coefficients" (Bocquet et al., 2013).

   Denote $\boldsymbol{w}_{\bullet}$ the control vector such that $\boldsymbol{x}(\boldsymbol{w}_{\bullet}) = \boldsymbol{x}_{\bullet}$, and note that $\boldsymbol{x}(\boldsymbol{e}_n) = \boldsymbol{x}_n$, where $\boldsymbol{e}_n$ is the $n$-th column of the identity matrix. Thus, $[\boldsymbol{x}_n - \boldsymbol{x}_{\bullet}] = \mathbf{X}[\boldsymbol{e}_n - \boldsymbol{w}_{\bullet}]$, and the prior increment (10a) with the ensemble estimates becomes:

$$\bar{\boldsymbol{\Delta}}_{\bullet}^{\mathrm{prior}} = (\mathbf{X} - \bar{\mathbf{K}}_i\mathbf{Y}_i)[\boldsymbol{e}_n - \boldsymbol{w}_{\bullet}], \tag{28}$$

where there is no explicit $\bar{\mathbf{M}}_i$, which only appears implicitly through $\mathbf{Y}_i = \bar{\mathbf{M}}_i\mathbf{X}$, as defined in equation (23) Alternatively, applying the subspace formula (24) and using $\mathbf{I}_N = \bar{\mathbf{C}}_{\boldsymbol{w},i}(\bar{\mathbf{C}}_{\boldsymbol{w},i})^{-1}$ yields:

$$\bar{\boldsymbol{\Delta}}_{\bullet}^{\mathrm{prior}} = \mathbf{X}\bar{\mathbf{C}}_{\boldsymbol{w},i}(N-1)[\boldsymbol{e}_n - \boldsymbol{w}_{\bullet}]. \tag{29}$$



## 3.5 Justifying the change of variables

**Lemma 1** (Closure). *Suppose $\mathbf{E}_i$ is generated by EnRML. Then, each member (column) of $\mathbf{E}_i$ is in the (prior) ensemble subspace. Moreover,* $\mathrm{col}(\mathbf{X}_i) \subseteq \mathrm{col}(\mathbf{X})$.

Lemma 1 may be proven by noting that $\mathbf{X}$ is the leftmost factor in $\bar{\mathbf{K}}_i$, and using induction on equations (10a) and (10b). Alternatively, it can be deduced (Raanes et al., 2018) as a consequence of the implicit assumption on the prior that $\boldsymbol{x} \sim \mathcal{N}(\bar{\boldsymbol{x}}, \bar{\mathbf{C}}_{\boldsymbol{x}})$. A stronger result, namely $\mathrm{col}(\mathbf{X}_i) = \mathrm{col}(\mathbf{X})$, is conjectured in appendix A, but Lemma 1 is sufficient for the present purposes: it implies that there exists $\boldsymbol{w}_\bullet \in \mathbb{R}^N$ such that $\boldsymbol{x}(\boldsymbol{w}_\bullet) = \boldsymbol{x}_\bullet$ for any ensemble member and any iteration. Thus, the lemma justifies the change of variables (27).

Moreover, using the ensemble control vector ($\boldsymbol{w}$) is theoretically advantageous as it inherently embodies the restriction to the ensemble subspace. A practical advantage is that $\boldsymbol{w}$ is relatively low-dimensional compared to $\boldsymbol{x}$, which lowers storage and accessing expenses.

## 3.6 Simplifying the regression

Recall the definition of equation (23): $\mathbf{Y}_i = \bar{\mathbf{M}}_i \mathbf{X}$. Avoiding the explicit computation of $\bar{\mathbf{M}}_i$ used in this product between the iteration-$i$ estimate $\bar{\mathbf{M}}_i$ and the initial (prior) $\mathbf{X}$ was the motivation behind the modification $\bar{\mathbf{C}}_{\boldsymbol{x}} \leftarrow \bar{\mathbf{C}}_{\boldsymbol{x},i}$ by Chen and Oliver (2013b). Here, instead, by simplifying the expression of the regression, it is shown how to compute $\mathbf{Y}_i$ without first computing $\bar{\mathbf{M}}_i$.

### 3.6.1 The transform matrix

Inserting the regression $\bar{\mathbf{M}}_i$ (19b) into the definition (23),

$$\mathbf{Y}_i = \mathcal{M}(\mathbf{E}_i)\, \mathbf{T}_i^+ , \tag{30}$$

where $\mathbf{T}_i^+ = \mathbf{X}_i^+ \mathbf{X}$ has been defined, apparently requiring the pseudo-inversion of $\mathbf{X}_i$ for each $i$. But, as shown in appendix A2,

$$\mathbf{T}_i = \mathbf{X}^+ \mathbf{X}_i , \tag{31}$$

which only requires the one-time pseudo-inversion of the prior anomalies, $\mathbf{X}$. Then, since the pseudo-inversion of $\mathbf{T}_i \in \mathbb{R}^{N \times N}$ for $\mathbf{Y}_i$ (30) is a relatively small calculation, this saves computational time.

The symbol $\mathbf{T}$ has been chosen in reference to deterministic, square-root EnKFs. Indeed, pre-multiplying equation (31) by $\mathbf{X}$ and recalling equation (17) and Lemma 1 produces $\mathbf{X}_i = \mathbf{X}\mathbf{T}_i$. Therefore, the "transform matrix", $\mathbf{T}_i$, describes the conditioning of the anomalies (and covariance).

Inversely, equation (30) can be seen as the "de-conditioning" of the posterior observation anomalies. This interpretation of $\mathbf{Y}_i$ should be contrasted to its definition (23), which presents it as the prior parameter anomalies "propagated" by the linearization of iteration $i$. The two approaches are known to be "mainly equivalent" in the deterministic case (Sakov et al., 2012). To our knowledge, however, it has not been exploited for EnRML before now, possibly because the proofs (appendix A2) are a little more complicated in this stochastic case.



### 3.6.2 From the ensemble controls

The ensemble matrix of iteration $i$ can be written:

$$\mathbf{E}_i = \bar{\boldsymbol{x}}\mathbb{1}^\mathsf{T} + \mathbf{X}\mathbf{W}_i, \tag{32}$$

where the columns of $\mathbf{W}_i \in \mathbb{R}^{N \times N}$ are the ensemble control vectors (27). Post-multiplying equation (32) by $\mathbf{\Pi}_\mathbb{1}^\perp$ to get the anomalies produces:

$$\mathbf{X}_i = \mathbf{X}(\mathbf{W}_i\mathbf{\Pi}_\mathbb{1}^\perp). \tag{33}$$

This seems to indicate that $\mathbf{W}_i\mathbf{\Pi}_\mathbb{1}^\perp$ is the transform matrix, $\mathbf{T}_i$, discussed in the previous subsection. However, they are not fully equal: inserting $\mathbf{X}_i$ from (33) into (31) yields:

$$\mathbf{T}_i = \mathbf{\Pi}_{\mathbf{X}^\mathsf{T}}(\mathbf{W}_i\mathbf{\Pi}_\mathbb{1}^\perp), \tag{34}$$

i.e. they are distinguished by $\mathbf{\Pi}_{\mathbf{X}^\mathsf{T}} = \mathbf{X}^+\mathbf{X}$: the projection onto the row space of $\mathbf{X}$.

Appendix A3 shows that, in most conditions, this pesky projection matrix vanishes when $\mathbf{T}_i$ is used in equation (30):

$$\mathbf{Y}_i = \mathcal{M}(\mathbf{E}_i)(\mathbf{W}_i\mathbf{\Pi}_\mathbb{1}^\perp)^+ \quad \text{if} \begin{cases} N-1 \le M, \text{ or} \\ \mathcal{M} \text{ is linear.} \end{cases} \tag{35}$$

In other words, the projection $\mathbf{\Pi}_{\mathbf{X}^\mathsf{T}}$ can be omitted unless $\mathcal{M}$ is nonlinear *and* the ensemble is larger than the unknown parameter's dimensionality.

A well known result of Reynolds et al. (2006) is that the first step of the EnRML algorithm (with $\mathbf{W}_0 = \mathbf{I}_N$) is equivalent to the EnKF. However, the standard definition of the EnKF uses cross-covariances rather than an explicit $\bar{\mathbf{M}}_0$ to define the

Kalman gain, and this corresponds to a $\mathbf{Y}_0$ that *never* contains $\mathbf{\Pi}_{\mathbf{X}^\mathsf{T}}$. The following section explains why it should be so for EnRML too.

### 3.6.3 Linearization chaining

Consider applying the change of variables (27) to $\boldsymbol{w}$ at the very beginning of the derivation of EnRML. Since $\mathbf{X}\mathbb{1} = 0$, there is a redundant degree of freedom in $\boldsymbol{w}$, meaning that there is a choice to be made in deriving its density from the original one,

given by $J_{\boldsymbol{x},n}(\boldsymbol{x})$ in equation (6). The simplest choice (Bocquet et al., 2015) results in the log-posterior:

$$J_{\boldsymbol{w},n}(\boldsymbol{w}) = \tfrac{1}{2}\|\boldsymbol{w} - \boldsymbol{e}_n\|^2_{\frac{1}{N-1}\mathbf{I}_N} + \tfrac{1}{2}\|\mathcal{M}(\bar{\boldsymbol{x}}+\mathbf{X}\boldsymbol{w}) - \boldsymbol{y}_n\|^2_{\mathbf{C}_\delta},$$

Application of the Gauss-Newton scheme with the gradients and Hessian of $J_{\boldsymbol{w},n}$, followed by a reversion to $\boldsymbol{x}$, produces the EnRML algorithm as developed above.

The derivation summarized in the previous paragraph is arguably simpler than that of the last few pages. Notably, (a) it

does not require the Woodbury identity to derive the subspace formulae; (b) there is never an explicit $\bar{\mathbf{M}}_i$ to deal with; (c) the





statistical linearization of least-squares regression from $\mathbf{W}_i$ to $\mathcal{M}(\mathbf{E}_i)$ directly yields equation (35), except that there are no preconditions.

While the case of a large ensemble ($N-1 > M$) is not typical in geoscience, the fact that this derivation does not produce a projection matrix (which requires a pseudo-inversion) under any conditions begs the questions: Why are they different? Which version is better?

The answers lie in understanding the linearization of the map $\boldsymbol{w} \mapsto \mathcal{M}(\bar{\boldsymbol{x}} + \mathbf{X}\boldsymbol{w})$, and noting that, similarly to analytical (infinitesimal) derivatives, the chain rule applies for least-squares regression. In effect, the product $\mathbf{Y}_i = \bar{\mathbf{M}}_i \mathbf{X}$, which potentially yields a projection matrix, can be seen as an application of the chain rule for the composite function $\mathcal{M}(\boldsymbol{x}(\boldsymbol{w}))$. By contrast, equation (35) – but without the precondition – is obtained by direct regression of the composite function. Typically, the two versions yield identical results (i.e. the chain rule). However, since the intermediate space, $\mathrm{col}(\mathbf{X})$, is of lower dimensions than the initial domain ($M < N-1$), indirect linear regression results in a loss of information, manifested by the projection matrix. Therefore, the definition $\mathbf{Y}_i = \mathcal{M}(\mathbf{E}_i)(\mathbf{W}_i \boldsymbol{\Pi}_{\mathbb{1}}^{\perp})^{+}$ is henceforth preferred to $\bar{\mathbf{M}}_i \mathbf{X}$.

Numerical experiments, as in section 4 but not shown, indicate no statistically significant advantage for either version. This corroborates similar findings by Sakov et al. (2012) for the deterministic flavour. Nevertheless, there is a practical advantage: avoiding the computation of $\boldsymbol{\Pi}_{\mathbf{X}^{\intercal}}$.

### 3.6.4 Inverting the transform

In square-root ensemble filters, the transform matrix should have $\mathbb{1}$ as an eigenvector (Sakov and Oke, 2008; Livings et al., 2008). By construction, this also holds true for $\mathbf{W}_i \boldsymbol{\Pi}_{\mathbb{1}}^{\perp}$, with eigenvalue 0. Now, consider adding $\mathbf{0} = \mathbf{X}\boldsymbol{\Pi}_{\mathbb{1}}$ to equation (33), yielding another valid transformation:

$$\mathbf{X}_i = \mathbf{X}\underbrace{(\mathbf{W}_i \boldsymbol{\Pi}_{\mathbb{1}}^{\perp} + \boldsymbol{\Pi}_{\mathbb{1}})}_{\boldsymbol{\Omega}_i}. \tag{36}$$

The matrix $\boldsymbol{\Omega}_i$, in contrast to $\mathbf{W}_i \boldsymbol{\Pi}_{\mathbb{1}}^{\perp}$ and $\mathbf{T}_i$, has eigenvalue 1 for $\mathbb{1}$, and can be shown to be invertible (Lemma 2, appendix A3). This is convenient for proving equation (35), as is done in appendix A3, where $\mathbf{Y}_i$ is initially expressed in terms of $\boldsymbol{\Omega}_i^{-1}$. Note, however, that this version requires centring $\mathcal{M}(\mathbf{E}_i)$ before post-multiplying by $\boldsymbol{\Omega}_i^{-1}$.

In real applications it is commonplace to use a stable linear solver in place of any inversion. Reflecting this, Algorithm 1 persists with $(\mathbf{W}_i \boldsymbol{\Pi}_{\mathbb{1}}^{\perp})^{+}$ rather than $\boldsymbol{\Omega}_i^{-1}$. However, in this pseudo-inversion, all $N-1$ non-zero singular values should be retained, and no truncation threshold should be used, because all components of $\mathbf{W}_i$ are equally important (unlike the old EnRML algorithm, where a $\mathbf{X}$ and/or $\mathbf{X}_i$ was scaled decomposed, and truncated). If, by extreme chance (cf. the conjecture in appendix A) combined with poor numerical precision or subroutines, the rank of $\mathbf{W}_i$ or its pseudo-inversion is lower, this will invalidate $J_{\boldsymbol{w},n}$ and the algorithm unless compensated for by pre-multiplying the prior increment on line 8 by that same projection.





### 3.7 Algorithm

To summarize, Algorithm 1 provides pseudo-code for the new EnRML formulation. The increments $\bar{\boldsymbol{\Delta}}^{\text{lklhd}}$ (26) and $\bar{\boldsymbol{\Delta}}^{\text{prior}}$ (29) can be recognized by pre-multiplying line 10 by $\mathbf{X}$. For aesthetics, the sign of the gradients has been reversed. Note that there is no need for an explicit iteration index. Nor is there an ensemble index, $n$, since all $N$ columns are stacked into the matrix $\mathbf{W}$.

65 However, in case $M$ is large, $\mathbf{Y}$ may be computed column-by-column to avoid storing $\mathbf{E}$. The product $\mathbf{W\Pi}_{\mathbb{1}}^{\perp}$ is computed by subtracting the column mean of $\mathbf{W}$. Its pseudo-inverse on line 6 should retain all $N-1$ non-zero singular values, as discussed in section 3.6.4. Line 9 may be computed using a reduced or truncated SVD of $\mathbf{C}_{\boldsymbol{\delta}}^{-1/2}\mathbf{Y}$, which is relatively fast for $N$ both larger and smaller than $P$. Alternatively, the Kalman gain forms could be used.

---

**Algorithm 1** Gauss-Newton variant of EnRML

(the stochastic flavour of the IEnKS analysis update)

**require:** prior ens. $\mathbf{E}$, obs. perturb's $\mathbf{D}$

1: $\bar{\boldsymbol{x}} \quad = \mathbf{E}\mathbb{1}/N$

2: $\mathbf{X} \quad = \mathbf{E} - \bar{\boldsymbol{x}}\mathbb{1}^{\mathsf{T}}$

3: $\mathbf{W} = \mathbf{I}_N$

4: **repeat:**

5:     Run model (on each col.) to get $\mathcal{M}(\mathbf{E})$

6:     $\mathbf{Y} \quad\quad = \mathcal{M}(\mathbf{E})\,(\mathbf{W\Pi}_{\mathbb{1}}^{\perp})^{+}$

7:     $\nabla J_{\mathbf{W}}^{\text{lklhd}} = \mathbf{Y}^{\mathsf{T}}\mathbf{C}_{\boldsymbol{\delta}}^{-1}[\boldsymbol{y}\mathbb{1}^{\mathsf{T}} + \mathbf{D} - \mathcal{M}(\mathbf{E})]$

8:     $\nabla J_{\mathbf{W}}^{\text{prior}} = (N-1)[\mathbf{I}_N - \mathbf{W}]$

9:     $\bar{\mathbf{C}}_{\boldsymbol{w}} \quad = \big(\mathbf{Y}^{\mathsf{T}}\mathbf{C}_{\boldsymbol{\delta}}^{-1}\mathbf{Y} + (N-1)\mathbf{I}_N\big)^{-1}$

10:     $\mathbf{W} \quad = \mathbf{W} + \bar{\mathbf{C}}_{\boldsymbol{w}}[\nabla J_{\mathbf{W}}^{\text{prior}} + \nabla J_{\mathbf{W}}^{\text{lklhd}}]$

11:     $\mathbf{E} \quad\quad = \bar{\boldsymbol{x}}\mathbb{1}^{\mathsf{T}} + \mathbf{XW}$

12: **until** tolerable convergence or max. iterations

13: **return** posterior ensemble $\mathbf{E}$

---

Localization may be implemented by local analysis. Tapering may be done by replacing the local-domain $\mathbf{C}_{\boldsymbol{\delta}}^{-1/2}$ by $\rho^{1/2} \circ$

70 $\mathbf{C}_{\boldsymbol{\delta}}^{-1/2}$, where $\circ$ is the Schur product, and $\rho$ is a square matrix containing the localization coefficients, $\rho_{m,l} \in [0,1]$. Also see Bocquet (2016); Chen and Oliver (2017) for localization of smoothers. Inflation and model error parameterizations are not included in the algorithm, but may be applied outside of it. Also see Sakov et al. (2017); Evensen (2018a) for model error treatment with iterative methods. The Levenberg-Marquardt variant is obtained by adding the trust-region parameter $\lambda > 0$ to $(N-1)$ in the Hessian, line 9, which impacts both the step length and direction.



## 4 Benchmark experiments

The new EnRML algorithm produces results that are *identical* to the old formulation, at least up to round-off and truncation errors, and for $N - 1 \leq M$. Therefore, since there is already a large number of studies of EnRML with reservoir cases (e.g., Chen and Oliver, 2013a; Emerick and Reynolds, 2013b), adding to this does not seem necessary.

However, there does not appear to be any studies of EnRML with the Lorenz-96 system (Lorenz, 1996) in a data assimilation setting. The advantages of this case are numerous: (a) the model is a surrogate of weather dynamics, and as such holds relevance in geoscience; (b) the problem is (exhaustively) sampled from the system's invariant measure, rather than being selected by the experimenter; (c) the sequential nature of data assimilation inherently tests prediction skill, which helps avoid the pitfalls of point measure assessment, such as overfitting; (d) its simplicity enhances reliability and reproducibility, and has made it a literature standard, thus facilitating comparative studies.

Comparison of the benchmark performance of EnRML will be made to the IEnKS, and ensemble multiple data assimilation (ES-MDA)[1], both its stochastic and deterministic (square-root) flavour. Not included in the benchmark comparisons is the version of EnRML where the prior increment is dropped (cf. section 1.1). This is because the chaotic, sequential nature of this case makes it practically impossible to achieve good results without propagating prior information. Similarly, as they lack a dynamic prior, this precludes "regularizing, iterative ensemble smoothers" (Iglesias, 2015), (Luo et al., 2015),[2] (Mandel et al., 2016)[3], even if their background is well-tuned, and their stopping condition judicious. Because they require the tangent-linear model, $\mathbf{M}_\bullet$, RML and EDA/En4DVar (Tian et al., 2008; Bonavita et al., 2012; Jardak and Talagrand, 2018) are not included. For simplicity, localization will not be used, nor covariance hybridization. Other, related methods may be found in the reviews of Bannister (2016); Carrassi et al. (2018).

### 4.1 Setup

The performances of the iterative ensemble smoother methods are benchmarked with "twin experiments", using the Lorenz-96 dynamical system, which is configured with standard settings (e.g., Ott et al., 2004; Bocquet and Sakov, 2014), detailed below. The dynamics are given by the $M = 40$ coupled ordinary differential equations:

$$\frac{\mathrm{d}x_m}{\mathrm{d}t} = (x_{m+1} - x_{m-2}) x_{m-1} - x_m + F, \tag{37}$$

for $m = 1, \ldots, M$, with periodic boundary conditions. These are integrated using the fourth-order Runge-Kutta scheme, with time steps of 0.05 time units, and no model noise. Observations of the entire state vector are taken $\Delta t_{\mathrm{obs}} = 0.2$ or $0.4$ time units apart with unit noise variance, meaning $\boldsymbol{y}(t) = \boldsymbol{x}(t) + \boldsymbol{\delta}(t)$, for each $t = k \cdot \Delta t_{\mathrm{obs}}$, with $k = 0, 1, \ldots, 10'000$, and $\mathbf{C}_{\boldsymbol{\delta}} = \mathbf{I}_M$.

---

[1]Note that this is MDA in the sense of Emerick and Reynolds (2013a), where the annealing itself yields iterations, and not in the sense of quasi-static assimilation (Pires et al., 1996; Bocquet and Sakov, 2014; Fillion et al., 2018), where it is used as an auxiliary technique.

[2]Their Lorenz-96 experiment only concerns the initial conditions.

[3]Their Lorenz-96 experiment seems to have failed completely, with most of the benchmark scores (their Figure 5) indicating divergence, which makes it pointless to compare benchmarks. Also, when reproducing their experiment, we obtain much lower scores than they report for the EnKF. One possible explanation is that we include, and tune, inflation.



The iterative smoothers are employed for the *filtering* problem, aiming to estimate $\boldsymbol{x}(t)$ as soon as $\boldsymbol{y}(t)$ comes in. In so doing, they also tackle the smoothing problem for $\boldsymbol{x}(t-L)$, where the data assimilation window has been fixed at $L = 0.4$, which is near optimal (cf. Bocquet et al., 2013, Figures 3 and 4). This window is shifted by $1 \times \Delta t_{\text{obs}}$ each time a new observation becomes available. A post-analysis inflation factor is tuned for optimal performance for each smoother and each ensemble size, $N$. Also, random rotations are used to generate the ensembles for the square-root variants. The number of iterations is fixed, either at 3 or 10. No tuning of the step length is undertaken: it is $1/3$ or $1/10$ for ES-MDA, and 1 for EnRML and the IEnKS.

The smoothers are assessed by their accuracy, as measured by root-mean squared error:

$$\text{RMSE}(t) = \sqrt{\frac{1}{M}\big\|\boldsymbol{x}(t) - \bar{\boldsymbol{x}}(t)\big\|_2^2}, \tag{38}$$

which is recorded immediately following each analysis of the latest observation $\boldsymbol{y}(t)$. After the experiment, the instantaneous $\text{RMSE}(t)$ are averaged for all $t > 20$. A table of RMSE averages is compiled for a range of $N$, and plotted as curves for each method, in Figure 1. All of the results can be reproduced using Python-code scripts hosted online at `https://github.com/nansencenter/DAPPER/tree/paper_StochIEnS`. This code reproduces previously published results in the literature. For example, our benchmarks obtained with the IEnKS can be cross-referenced with the ones reported by Bocquet and Sakov (2014, Figure 7a).

## 4.2 Results

As we expect, Figure 1 shows that the performance of all of the smoothers improve with increasing $N$, which needs to be at least 15 for tolerable performance, corresponding to the rank of the unstable subspace of the dynamics plus one (Bocquet and Carrassi, 2017). Of course, all of the scores are lower for the left pane where $\Delta t_{\text{obs}} = 0.2$, compared to the right pane where $\Delta t_{\text{obs}} = 0.4$.

The deterministic (square-root) IEnKS and ES-MDA score noticeably lower RMSE averages than the stochastic IEnKS (i.e. EnRML) and ES-MDA, which require $N$ closer to 30 for tolerable performance. Also tested (not shown) was the first-order-approximate deterministic flavour of ES-MDA (Emerick, 2018), which systematically performed slightly worse than the square-root flavour.

It appears that 3 iterations is largely sufficient, since its markers are rarely significantly higher than those of 10 iterations, the exceptions all occurring when the ensemble size is close to the lower limit of the tolerable performance range.

Between the two stochastic smoothers (EnRML and stochastic ES-MDA) there is no clear-cut advantage. Among the deterministic smoothers, the IEnKS performs slightly better than ES-MDA, though this is hardly significant. This result came as a surprise because, in contrast with EnRML/IEnKS which can iterate indefinitely, we thought that ES-MDA would suffer from occasionally not "reaching" the optimum.

One explanation could be that EnRML/IEnKS need a lowering of the step lengths, possibly as a function of the iteration number, to avoid causing "unphysical" states, and to avoid "bouncing around" near the optimum. Along with the related MDA-inflation parameter, tuning of the step length has been subject of several studies (Chen and Oliver, 2012; Bocquet and Sakov,





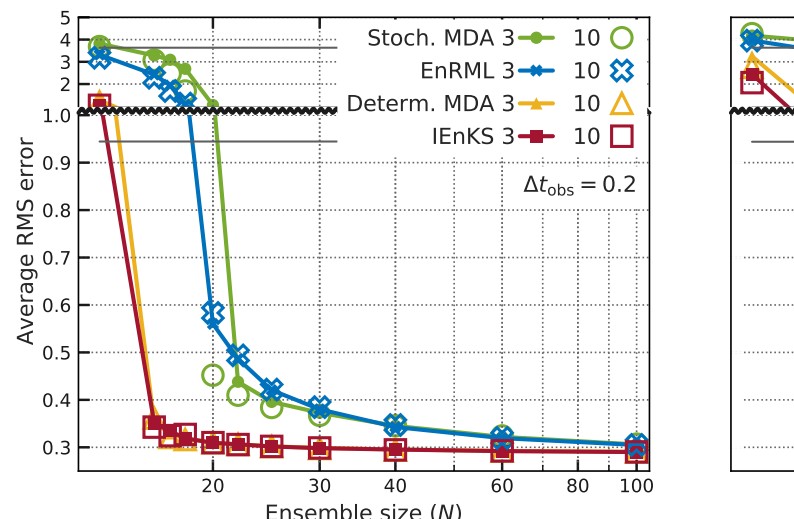 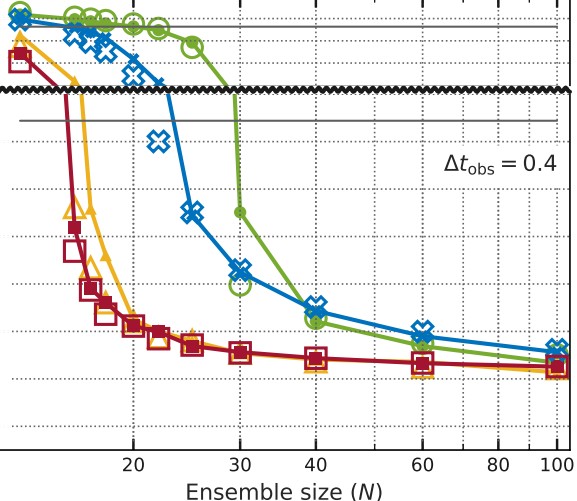

**Figure 1.** Benchmarks of the (filtering) accuracy of four iterative ensemble smoothers, obtained with the Lorenz-96 system, plotted as functions of $N$. The $y$-axis changes resolution at $y = 1$. For perspective, the two lines at $y = 3.6$ and $y = 0.94$ show the average RMSE of the climatological mean, and of the optimal interpolation method, respectively. Each of the iterative ensemble smoothers is plotted for 3 (compact markers) and 10 (hollow markers) iterations. It can be seen that the deterministic (i.e. square-root) methods systematically achieve lower RMSE averages.

2012; Ma et al., 2017; Le et al., 2016; Rafiee and Reynolds, 2017). However, our superficial trials with this parameter (not shown) yielded little or no improvement.

## 45  5  Summary

This paper has presented a new and simpler (on paper and computationally) formulation of the iterative, stochastic ensemble smoother known as ensemble randomized maximum likelihood (EnRML). Notably, there is no explicit computation of the sensitivity matrix $\bar{\mathbf{M}}_i$, while the product $\mathbf{Y}_i = \bar{\mathbf{M}}_i\mathbf{X}$ is computed without any pseudo-inversions of the matrix of parameter anomalies. This fixes issues of noise, computational cost, and covariance localization, and there is no longer any temptation 50 to omit the prior increment from the update. Moreover, the Levenberg-Marquardt variant is now a trivial modification of the Gauss-Newton variant.

The new EnRML formulation was obtained by improvements to the background theory and derivation. Notably, Theorem 1 established the relation of the ensemble-estimated, least-squares linear regression coefficients, $\bar{\mathbf{M}}_i$, to "average sensitivity". Section 3.6 then showed that the computation of its action on the prior anomalies, $\mathbf{Y}_i = \bar{\mathbf{M}}_i\mathbf{X}$, simplifies into a de-conditioning 55 transformation, $\mathbf{Y}_i = \mathcal{M}(\mathbf{E}_i)\mathbf{T}_i^+$. Further computational gains resulted from expressing $\mathbf{T}_i$ in terms of the control vectors, $\mathbf{W}_i$, except that it also involves the "annoying" $\mathbf{\Pi}_{\mathbf{X}^\top}$. Although it usually vanishes, the appearance of this projection is likely





the reason why most expositions of the EnKF do not go the length of declaring that its implicit linearization of $\mathcal{M}$ is that of least-squares linear regression. Section 3.6.3 showed that the projection is merely the result of using the chain rule for indirect regression to the ensemble space, and argued that it is preferable to use the direct regression of the standard EnKF.

The other focus of the derivation was rank issues, with $\bar{\mathbf{C}}_{\boldsymbol{x}}$ not assumed invertible. Using the Woodbury matrix lemma, and avoiding implicit pseudo-inversions and premature insertion of SVDs, it was shown that the rank deficiency invalidates the Hessian form of the RML update, which should be restricted to the ensemble subspace. On the other hand, the subspace form and Kalman gain form of the update remain equivalent and valid. Furthermore, Theorem 2 of appendix A prove that the ensemble does not lose rank during the updates of EnRML (or EnKF).

The paper has also drawn significantly on the theory of the deterministic counterpart to EnRML: the iterative ensemble Kalman smoother (IEnKS). Comparative benchmarks using the Lorenz-96 model with these two and the ensemble multiple data assimilation (ES-MDA) smoother were shown in section 4. As in the non-iterative case (e.g., Sakov and Oke, 2008), the deterministic smoothers achieved better accuracy than the stochastic methods. Surprisingly, there was is little performance difference between ES-MDA and EnRML/IEnKS.

**Appendix A: Proofs**

**A1   Preliminary**

*Proof of Theorem 1.*   Assume $0 < |\mathbf{C}_{\boldsymbol{x}}| < \infty$, and that each element of $\mathbf{C}_{\mathcal{M}(\boldsymbol{x}),\boldsymbol{x}}$ and $\mathbb{E}[\mathcal{M}'(\boldsymbol{x})]$ is finite. Then $\bar{\mathbf{C}}_{\boldsymbol{x}}$ is a strongly consistent estimator of $\mathbf{C}_{\boldsymbol{x}}$. Likewise, $\bar{\mathbf{C}}_{\mathcal{M}(\boldsymbol{x}),\boldsymbol{x}} \to \mathbf{C}_{\mathcal{M}(\boldsymbol{x}),\boldsymbol{x}}$ almost surely, as $N \to \infty$. Thus, since $\bar{\mathbf{M}} = \bar{\mathbf{C}}_{\mathcal{M}(\boldsymbol{x}),\boldsymbol{x}} \bar{\mathbf{C}}_{\boldsymbol{x}}^{-1}$ for sufficiently large $N$, Slutsky's theorem yields $\bar{\mathbf{M}} \to \mathbf{C}_{\mathcal{M}(\boldsymbol{x}),\boldsymbol{x}} \mathbf{C}_{\boldsymbol{x}}^{-1}$, almost surely. The equality to $\mathbb{E}[\mathcal{M}'(\boldsymbol{x})]$ follows directly
from "Stein's lemma" (Liu, 1994).   □

**Theorem 2** (EnKF rank preservation). *The posterior ensemble's covariance, obtained using the EnKF, has the same rank as the prior's, almost surely (a.s.).*

*Proof.*   The updated anomalies, both for the square-root and the stochastic EnKF, can be written $\mathbf{X}^a = \mathbf{X}\mathbf{T}^a$ for some $\mathbf{T}^a \in \mathbb{R}^{N \times N}$.

For a deterministic EnKF, $\mathbf{T}^a = \sqrt{N-1}\bar{\mathbf{C}}_{\boldsymbol{w}}^{-1/2}$ for some matrix square-root (Sakov and Oke, 2008). Indeed, $\bar{\mathbf{C}}_{\boldsymbol{w}}$ is symmetric, positive, definite, and full-rank. Hence $\mathrm{rank}(\mathbf{X}^a) = \mathrm{rank}(\mathbf{X})$.

For the stochastic EnKF, equations (24) and (26) may be used to show that $\mathbf{T}^a = (N-1)\bar{\mathbf{C}}_{\boldsymbol{w}}\boldsymbol{\Upsilon}\boldsymbol{\Pi}_{\mathbb{1}}^{\perp}$, with $\boldsymbol{\Upsilon} = \mathbf{I}_N + \mathbf{Y}^{\top}\mathbf{C}_{\boldsymbol{\delta}}^{-1}\mathbf{D}/(N-1)$. Hence, for rank preservation, it will suffice to show that $\boldsymbol{\Upsilon}$ is a.s. full rank.

We begin by writing $\boldsymbol{\Upsilon}$ more compactly:

$$\boldsymbol{\Upsilon} = \mathbf{I}_N + \mathbf{S}^{\top}\mathbf{Z} \quad \text{with} \begin{cases} \mathbf{S} = (N-1)^{-1/2}\mathbf{C}_{\boldsymbol{\delta}}^{-1/2}\mathbf{Y}, \\ \mathbf{Z} = (N-1)^{-1/2}\mathbf{C}_{\boldsymbol{\delta}}^{-1/2}\mathbf{D}. \end{cases} \tag{A1}$$





From equations (4), (14) and (A1) it can be seen that column $n$ of $\mathbf{Z}$ follows the law $\boldsymbol{z}_n \sim \mathcal{N}(\mathbf{0}, \mathbf{I}_P/(N-1))$. Hence, column $n$ of $\boldsymbol{\Upsilon}$ follows $\boldsymbol{v}_n \sim \mathcal{N}(\boldsymbol{e}_n, \mathbf{S}^\mathsf{T}\mathbf{S}/(N-1))$, and has sample space:

$$\mathcal{S}_n = \{\boldsymbol{v} \in \mathbb{R}^N \ : \ \boldsymbol{v} = \boldsymbol{e}_n + \mathbf{S}^\mathsf{T}\boldsymbol{z}\}. \tag{A2}$$

Now consider, for $n = 0, \ldots, N$, the hypothesis:

$$\operatorname{rank}([\boldsymbol{\Upsilon}_{:n},\ \mathbf{I}_{n:}]) = N, \tag{$\mathrm{H}_n$}$$

where $\boldsymbol{\Upsilon}_{:n}$ denotes the first $n$ columns of $\boldsymbol{\Upsilon}$, and $\mathbf{I}_{n:}$ denotes the last $N-n$ columns of $\mathbf{I}_N$. Clearly, $\mathrm{H}_0$ is true. Now, suppose $\mathrm{H}_{n-1}$ is true. Then the columns of $[\boldsymbol{\Upsilon}_{:n-1},\ \mathbf{I}_{n-1:}]$ are all linearly independent. For column $n$, this means that $\boldsymbol{e}_n \notin \operatorname{col}([\boldsymbol{\Upsilon}_{:n-1},\ \mathbf{I}_{n:}])$. By contrast, from equation (A2), $\boldsymbol{e}_n \in \mathcal{S}_n$. The existence of a point in $\mathcal{S}_n \setminus \operatorname{col}([\boldsymbol{\Upsilon}_{:n-1},\ \mathbf{I}_{n:}])$ means that

$$\dim\big(\mathcal{S}_n \cap \operatorname{col}([\boldsymbol{\Upsilon}_{:n-1},\ \mathbf{I}_{n:}])\big) < \dim(\mathcal{S}_n). \tag{A3}$$

Since $\boldsymbol{v}_n$ is absolutely continuous with sampling space $\mathcal{S}_n$, equation (A3) means that the probability that $\boldsymbol{v}_n \in \operatorname{col}([\boldsymbol{\Upsilon}_{:n-1},\ \mathbf{I}_{n:}])$ is zero. This implies $\mathrm{H}_n$ a.s., establishing the induction. Identifying the final hypothesis $(\mathrm{H}_N)$ with $\operatorname{rank}(\boldsymbol{\Upsilon}) = N$ concludes the proof for the EnKF. □

A corollary of Theorem 2 and Lemma 1 is that the ensemble subspace is also unchanged by the EnKF update. Note that both the prior ensemble and the model (involved through $\mathbf{Y}$) are arbitrary in Theorem 2. However, $\mathbf{C}_{\boldsymbol{\delta}}$ is assumed invertible. The result is therefore quite different from the topic discussed by Kepert (2004); Evensen (2004), where rank deficiency arises due to a reduced-rank $\mathbf{C}_{\boldsymbol{\delta}}$.

**Conjecture 1.** *The rank of the ensemble is preserved by the EnRML update (a.s.) and* $\mathbf{W}_i$ *is invertible.*

We were not able to prove Conjecture 1, but it seems a logical extension of Theorem 2, and is supported by numerical trials. The following proofs utilize Conjecture 1, without which some projections will not vanish. Yet, even if Conjecture 1 should not hold (due to bugs, truncation, or really bad luck), Algorithm 1 is still valid and optimal, as discussed in sections 3.6.3 and 3.6.4.

### A2 The transform matrix

**Theorem 3.** $(\mathbf{X}^+\mathbf{X}_i)^+ = \mathbf{X}_i^+\mathbf{X}$.

*Proof.* Let $\mathbf{T} = \mathbf{X}^+\mathbf{X}_i$ and $\mathbf{S} = \mathbf{X}_i^+\mathbf{X}$. The following shows that $\mathbf{S}$ satisfies the four properties of the Moore-Penrose characterization of the pseudo-inverse of $\mathbf{T}$:

1. $\mathbf{TST} = (\mathbf{X}^+\mathbf{X}_i)(\mathbf{X}_i^+\mathbf{X})(\mathbf{X}^+\mathbf{X}_i)$

$$= \mathbf{X}^+\boldsymbol{\Pi}_{\mathbf{X}_i}\boldsymbol{\Pi}_{\mathbf{X}}\mathbf{X}_i \qquad [\boldsymbol{\Pi}_{\mathbf{A}} = \mathbf{A}\mathbf{A}^+]$$

$$= \mathbf{X}^+\boldsymbol{\Pi}_{\mathbf{X}_i}\mathbf{X}_i \qquad\qquad [\text{Lemma 1}]$$

$$= \mathbf{T}. \qquad\qquad\qquad [\boldsymbol{\Pi}_{\mathbf{A}}\mathbf{A} = \mathbf{A}]$$





2. $\mathbf{STS} = \mathbf{S}$, as may be shown similarly to point 1.

3. $\mathbf{TS} = \mathbf{X}^+\mathbf{X}$, as may be shown similarly to point 1, using Conjecture 1. The symmetry of $\mathbf{TS}$ follows from that of $\mathbf{X}^+\mathbf{X}$.

4. The symmetry of $\mathbf{ST}$ is shown as for point 3. □

This proof was heavily inspired by appendix A of Sakov et al. (2012). However, our developments apply for EnRML (rather than the deterministic, square-root IEnKS). This means that $\mathbf{T}_i$ is not symmetric, which complicates the proof in that the focus must be on $\mathbf{X}^+\mathbf{X}_i$ rather than $\mathbf{X}_i^+$ alone. Our result also shows the equivalence of $\mathbf{S}^+$ and $\mathbf{T}$ in general, while the additional result of the vanishing projection matrix in the case of $N-1 \le M$ is treated as a corollary, shown in the following.

## A3  Proof of equation (35)

**Lemma 2.** $\boldsymbol{\Omega}_i$ *is invertible (provided* $\mathbf{W}_i$ *is).*

*Proof.* We show that $\boldsymbol{\Omega}_i\boldsymbol{u} \ne 0$ for any $\boldsymbol{u} \ne 0$, where $\boldsymbol{\Omega}_i = \mathbf{W}_i\boldsymbol{\Pi}_{\mathbb{1}}^{\perp} + \boldsymbol{\Pi}_{\mathbb{1}}$. For $\boldsymbol{u} \in \mathrm{col}(\mathbb{1})$: $\boldsymbol{\Omega}_i\boldsymbol{u} = \boldsymbol{u}$. For $\boldsymbol{u} \in \mathrm{col}(\mathbb{1})^{\perp}$: $\boldsymbol{\Omega}_i\boldsymbol{u} = \mathbf{W}_i\boldsymbol{u} \ne 0$ (Conjecture 1). □

Recall that equation (34) was obtained by inserting $\mathbf{X}_i$ in the expression (31) for $\mathbf{T}_i$. The following uses the alternative of inserting $\mathbf{X}$ in the expression (30) for $\mathbf{T}_i^+$.

By equation (36) and Lemma 2, $\mathbf{X} = \mathbf{X}_i\boldsymbol{\Omega}_i^{-1}$ and so $\mathbf{T}_i^+ = \boldsymbol{\Pi}_{\mathbf{X}_i^{\mathsf{T}}}\boldsymbol{\Omega}_i^{-1}$. We now re-introduce $\boldsymbol{\Pi}_{\mathbb{1}}^{\perp}$, which was omitted for equation (19b), by prepending it to $\mathbf{T}_i^+$; this does not change its value. In summary, equation (30) becomes:

$$\mathbf{Y}_i = [\mathcal{M}(\mathbf{E}_i)\boldsymbol{\Pi}_{\mathbb{1}}^{\perp}]\boldsymbol{\Pi}_{\mathbf{X}_i^{\mathsf{T}}}\boldsymbol{\Omega}_i^{-1}. \tag{A4}$$

Next, it is shown that, under certain conditions, the projection matrix $\boldsymbol{\Pi}_{\mathbf{X}_i^{\mathsf{T}}}$ vanishes:

$$\mathbf{Y}_i = [\mathcal{M}(\mathbf{E}_i)\boldsymbol{\Pi}_{\mathbb{1}}^{\perp}]\boldsymbol{\Omega}_i^{-1}. \tag{A5}$$

Thereafter, appendix A4 can be used to write $\boldsymbol{\Omega}_i^{-1}$ in terms of $(\mathbf{W}_i\boldsymbol{\Pi}_{\mathbb{1}}^{\perp})^+$.

**The case of $N-1 \le M$**

In the case of $N-1 \le M$, the null space of $\mathbf{X}$ is the range of $\mathbb{1}$ (with probability 1, Muirhead, 1982, Theorem 3.1.4). By Lemma 2, the same applies for $\mathbf{X}_i$, and so $\boldsymbol{\Pi}_{\mathbf{X}_i^{\mathsf{T}}}$ in equation (A4) reduces to $\boldsymbol{\Pi}_{\mathbb{1}}^{\perp}$. □

**The case of linearity**

Let $\mathbf{M}$ be the matrix of the observation model $\mathcal{M}$, here assumed linear: $\mathcal{M}(\mathbf{E}_i) = \mathbf{M}\mathbf{E}_i$. By equation (A4), $\mathbf{Y}_i = \mathbf{M}\mathbf{E}_i\boldsymbol{\Pi}_{\mathbf{X}_i^{\mathsf{T}}}\boldsymbol{\Omega}_i^{-1}$. But $\mathbf{E}_i\boldsymbol{\Pi}_{\mathbf{X}_i^{\mathsf{T}}} = \mathbf{X}_i = \mathbf{E}_i\boldsymbol{\Pi}_{\mathbb{1}}^{\perp}$. □



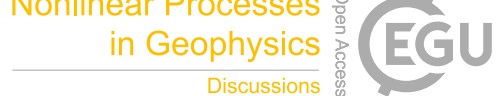

## A4  The pseudo-inverse version

The results of this section do not depend on whether the projection $\mathbf{\Pi}_{\mathbf{X}}$ is included in $\mathbf{Y}_i$ or not. Either way, $\mathbf{Y}_i\mathbb{1} = 0$, and so

$$\bar{\mathbf{C}}_{\boldsymbol{w},i}^{-1}\mathbb{1} = \mathbb{1} = \bar{\mathbf{C}}_{\boldsymbol{w},i}\mathbb{1}\,, \tag{A6}$$

where $\bar{\mathbf{C}}_{\boldsymbol{w},i}$ is defined in equation (25), and the second equality follows from the first. Similarly, the following identities are valid also when $\mathbf{W}_i$ and $\mathbf{W}_i^{-1}$ are swapped.

$$\mathbf{W}_i^\mathsf{T}\mathbb{1} = \mathbb{1}\,; \tag{A7}$$
$$\mathbf{W}_i\mathbf{\Pi}_{\mathbb{1}}^{\perp} = \mathbf{\Pi}_{\mathbb{1}}^{\perp}\mathbf{W}_i\mathbf{\Pi}_{\mathbb{1}}^{\perp}\,; \tag{A8}$$
$$(\mathbf{W}_i\mathbf{\Pi}_{\mathbb{1}}^{\perp})^+ = \mathbf{\Pi}_{\mathbb{1}}^{\perp}\mathbf{W}_i^{-1}\mathbf{\Pi}_{\mathbb{1}}^{\perp}\,. \tag{A9}$$

Equation (A7) is proven inductively (in $i$) using (A6) in line 10 of Algorithm 1. It enables showing (A8), using $\mathbf{\Pi}_{\mathbb{1}}^{\perp} = \mathbf{I}_N - \mathbf{\Pi}_{\mathbb{1}}$. This enables showing (A9), similarly to Theorem 3. These identities can then be used to verify (by multiplying with $\mathbf{\Omega}_i$) that

$$\mathbf{\Omega}_i^{-1} = (\mathbf{W}_i\mathbf{\Pi}_{\mathbb{1}}^{\perp})^+ + \mathbf{\Pi}_{\mathbb{1}}\,. \tag{A10}$$

Substituting this formula for $\mathbf{\Omega}_i^{-1}$ into equation (A5) then reduces it to the pseudo-inverse version (35). As for equation (19b), the projection $\mathbf{\Pi}_{\mathbb{1}}^{\perp}$ can again be omitted.

*Author contributions.* The new and simpler EnRML algorithm was derived by PNR and further developed in consultation with GE (who also developed much of it independently) and ASS. Theorems 1 and 2 were derived by PNR, prompted by discussions with GE, and verified by ASS. The experiments and the rest of the writing were done by PNR and revised by GE and ASS.

*Competing interests.* The authors declare that they have no conflict of interest

*Acknowledgements.* The authors thank Dean Oliver, Kristian Fossum and Marc Bocquet for their pointers about the wider literature, and Elvar Bjarkason for his question about the computation of $(\mathbf{W}_i\mathbf{\Pi}_{\mathbb{1}}^{\perp})^+$. This work has been funded by DIGIRES, a project sponsored by industry partners and the PETROMAKS2 programme of the Research Council of Norway.



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
