# Peer review of "Revising the stochastic iterative ensemble smoother"

_Nonlinear Processes in Geophysics, 2019_

## Referee Comment (RC1) · Pavel Sakov (Referee) · 11 Apr 2019

[11pt]article natbib amsmath amssymb [breaklinks]hyperref xcolor

[Figure]

**Review of manuscript NPG-2019-10 "Revising the stochastic iterative ensemble smoother" by Patrick N. Raanes, Andreas S. Stordal, and Geir Evensen**

Pavel Sakov

April 11, 2019

This is a useful study that nicely sums up the existing literature on the stochastic EnKS (mainly known as EnRML), offers a simplified and more efficient algorithm, and compares performance of the method with that of a few other popular iterative schemes in two Lorenz-96 based experiments.

Apart from a few suggestions and questions below, the manuscript overall seems to be in a good shape to recommend it to **publish with a minor revision**.

1. P.1, L.15: **IES**

   The Ensemble Smoother (ES) van Leeuwen and Evensen (1996) is a non-sequential method. It involves no ensemble update during the whole run of the system. While ES produces an optimal analysis for linear systems only, it also seems to be a popular approach for nonlinear systems with stable dynamics, e.g. in reservoir modelling.

Interactive
comment

In contrast, the EnRML is a sequential method specifically designed for nonlinear dynamics. I am therefore a bit lost when the authors say that EnRML is "also known as the iterative ensemble smoother (IES)", particularly considering that the paper in general is fairly thorough in reviewing the literature, and that Geir Evensen is one of the authors.

If the above is correct, then using the term "ES-MDA" in regard to a sequential method also seems somewhat controversial.

2. P.1., L.18 and further: **Gu et al.**

   Should be "Gu and Oliver".

3. P.5, L.49: $\mathbf{C}_y$ **may be identified as the prior covariance of observation**

   Should be "as the prior covariance of innovation".

4. P.5, L.61: **Sakov et al. (2017)**

   Should be "Sakov et al. (2018)".

5. P.7, L.26: **Using the $\mathbf{C}_x$ in its stead is flawed and damaging because it is zero in the directions orthogonal to the ensemble subspace, so that its use would imply that the prior is assumed infinitely uncertain (i.e. flat) as opposed to infinitely certain (like a delta function) in those directions.**

   I used to believe that using $\mathbf{C}_x^+$ is not a problem because the cost function is estimated in the ensemble subspace only.

6. P.8, L.57: **Bocquet et al.,2013**

   Should be "Bocquet and Sakov, 2013".

7. P.14, L.12: **where the data assimilation window has been fixed at L = 0.4, which is near optimal (cf. Bocquet et al., 2013, Figures 3 and 4)**

   I am missing the point why $L = 0.4$ is near optimal.

8. Section 4.2

   It would also be interesting to see results for a more nonlinear case $\Delta t_{obs} = 0.8$.

**References**

van Leeuwen, P. J. and G. Evensen, 1996: Data assimilation and inverse methods in terms of a probabilistic formulation. *Mon. Wea. Rev.*, **124**, 2898–2913.

---

## Referee Comment (RC2) · Anonymous Referee #2 · 14 Apr 2019

This manuscript reformulates the algorithm of the original ensemble randomized maximum likelihood (EnRML), a stochastic EnKS, and compares its performance with other iterative schemes using the Lorenz-96 model. The manuscript is well written. I basically have no critical comment on the proposed method. However, I would like to recommend the authors performing some additional Lorenz-96 experiments to discuss differences between EnRML and iterative EnKS. For example, the following experiments would be interesting:

(1) experiments with imperfect observation error variance

(2) experiments with higher nonlinearily (e.g., larger obs error or infrequent assimilation)

(3) experiments with model imperfection

Those additional experiments, I believe, should improve the manuscript more.

[Minor comments]

1. Please add dimensions of matrices as Eqs. (13) and (14) so that we can follow equations easier.

2. I recommend the authors to add a schematic figure that simply illustrates the proposed method compared with the original EnRML and iterative EnKS.

3. "stochastic and deterministic ES-MDA" should be explained more.
* * *

---

## Referee Comment (RC3) · Marc Bocquet (Referee) · 5 May 2019

This is a very nice paper. Taking inspiration from the deterministic IEnKS, it significantly clarifies the derivation of the EnRML which I personally always found a bit convoluted. In spite of all the technical details, it is a relief to see the simple and sleek algorithm in page 12 of the manuscript. It allows for an immediate comparison with the deterministic IEnKS and it makes sense even without going into the details of the derivation. Part (and only part) of the simplification is reminiscent of the simplification operated in Bocquet and Sakov (2012) onto the IEnKF algorithm of Sakov et al. (2012) (line 21 of the main algorithm in Appendix A) without fundamentally changing the method. This could be briefly mentioned. I have not checked the appendices which are very technical and would require too much time to confidently check.

You will find below a list of minor suggestions, which could help improve the manuscript:

1. The line numbering is inconsistent (unpractical at best)!

2. line 1: The parentheses around stochastic are unnecessary.

3. line 12: "reservoir" → "oil reservoir": I believe reservoir is not natural for the usual NPG reader.

4. line 16: Readers unfamiliar with the EnRML may jump to the beginning of the derivation: section 2: But then, there would be no introduction at all for theses readers. They do deserve a few sentences! (the abstract does not count.)

5. lines 27-28: Parentheses are missing around the equations' number.

6. line 44: Bocquet and Sakov (2012) could be mentioned here, as it showed that the requirement for inverses in Sakov et al. (2012) was unnecessary.

7. lines 50-52: I guess you are missing the key point which is that chaotic model require sequential assimilation, whereas oil reservoir models, although nonlinear (and non trivial), are not chaotic.

8. line 59: "some of the results" → "some of the mathematical results" (otherwise, the statement would seem odd).

9. line 64: This has also been discussed in Liu et al. (2017) and to some extent Morzfeld et al. (2018).

10. page 3, line 12: Please define $\|$.

11. page 4, line 19: The section title is too generic and not consistent with the one chosen for section 2.3. "RML" would be more consistent.

12. page 4, line 30: "first-order" is ambiguous. Strictly speaking, this is second-order as a Taylor expansion but without second-order derivatives.

13. page 4, line 41: Please define $\mathbf{I}_M$.

14. Page 5, line 49: $\mathbf{C}_y$ is often called the innovation covariance matrix, or innovation statistics.

15. Page 5, line 50: "than the inversion to compute $\mathbf{C}$" → "than the inversion needed to compute $\mathbf{C}_\bullet$ in (7b)."

16. page 5, line 62: "yield" → "yields".

17. page 5, line 68: "without the costly use of Metropolis-Hastings": More fundamentally this is due to the curse of dimensionality (see e.g., Liu et al. (2017)).

18. page 7, line 6: "Maciejewski and Klein, 1985" → "as proven by ..." or use parentheses.

19. page 7, Theorem 1: At the NPG level, this result can be seen and written down on the back of an envelope, and that is why nobody cares to do so. It is nice though that you show it rigorously but since the heuristic for this proof is adamant, this would not really be required in NPG. By that I mean that previous authors are not really guilty. Also, I believe that the heuristic derivation is not restricted to Gaussian distributions, is it?

20. page 8, lines 41-48: This can be better understood thinking in terms of conditioning of the associated cost function.

21. page 8, line 58: "the control vector" → "a control vector" right? there may be several.

22. page 9, line 90: "Inversely" → "Conversely".

23. page 12, line 63: "pre-multiplying" is ambiguous. Say for instance on the right or on the left.

24. page 13, line 85: There is also a quasi-static variant of the deterministic IEnKS that would be worth considering (Fillion et al., 2018). However the data assimilation window length your chose might not be long enough to do so.

25. page 13, line 10: 10'000, do you mean 10,000?

26. page 14, line 18: I would add the smoothing RMSE to the numerical study to muscle it up. Moreover, I noticed that it is often uneasy for colleagues to grasp that such filter yields better filtering RMSE. I would show both the filtering and smoothing RMSEs to tell that these methods do both with high accuracy.

27. page 14, line 31: You could mention that this is qualitatively if not quantitatively similar to the deterministic versus stochastic EnKF.

28. Page 14, line 42: "has been subject of" → "has been the subject of".

29. page 14, line 42: "tuning of the step length": what does it mean? I really have no clue.

30. page 15, line 48: This is really reminiscent of the improvement brought about by Bocquet and Sakov (2012) onto Sakov et al. (2012) in the context of the deterministic IEnKF/IEnKS.

31. page 16, line 57: "do not go the length of" → "do not go to the length of".

32. page 16, line 63: "prove" → "proves".

33. page 16, line 68: "was is" → "was".

**References**

Bocquet, M., Sakov, P., 2012. Combining inflation-free and iterative ensemble Kalman filters for strongly nonlinear systems. Nonlin. Processes Geophys. 19, 383–399. doi:`10.5194/npg-19-383-2012`.

Fillion, A., Bocquet, M., Gratton, S., 2018. Quasi static ensemble variational data assimilation: a theoretical and numerical study with the iterative ensemble Kalman smoother. Nonlin. Processes Geophys. 25, 315–334. doi:`10.5194/npg-25-315-2018`.

Liu, Y., Haussaire, J.M., Bocquet, M., Roustan, Y., Saunier, O., Mathieu, A., 2017. Uncertainty quantification of pollutant source retrieval: comparison of Bayesian methods with application to the Chernobyl and Fukushima-Daiichi accidental releases of radionuclides. Q. J. R. Meteorol. Soc. 143, 2886–2901. doi:`10.1002/qj.3138`.

Morzfeld, M., Hodyss, D., Poterjoy, J., 2018. Variational particle smoothers and their localization. Q. J. R. Meteorol. Soc. 144, 806–825. doi:`10.1002/qj.3256`.

Sakov, P., Oliver, D.S., Bertino, L., 2012. An iterative EnKF for strongly nonlinear systems. Mon. Wea. Rev. 140, 1988–2004. doi:`10.1175/MWR-D-11-00176.1`.

---

## Author Comment (AC1) · 3 Jul 2019

The attached supplement (zip) contains:

Manuscript2.pdf + Response.pdf + diff.pdf

Please also note the supplement to this comment:
https://www.nonlin-processes-geophys-discuss.net/npg-2019-10/npg-2019-10-AC1-supplement.zip

---

## Author Response (AR1)

**Response to review comments for manuscript NPG-2019-10:**
**Revising the stochastic iterative ensemble smoother**

Patrick N. Raanes, Geir Evensen, Andreas S. Stordal

July 3, 2019

We thank editors and reviewers for their work; the feedback has been very useful. Pointwise replies to the review comments can be seen below. We hope the present version will satisfy the high standards of the journal of nonlinear processes in geophysics.

A latex-diff pdf file is attached, tracking the changes made. These are largely of a minor character. The most significant changes (prompted by reviews) are (i) the addition of some experiments (ii) moving the "layout" section into section 1.0, and (iii) improved clarity around Theorem 1. Additionally, (iv) details have been added around Algorithm 1, and and (v) we have changed the recommended form of the inverse transform matrix (and done cosmetic improvements to the accompanying appendix).

**RC1 (Pavel Sakov)**

This is a useful study that nicely sums up the existing literature on the stochastic EnKS (mainly known as EnRML), offers a simplified and more efficient algorithm, and compares performance of the method with that of a few other popular iterative schemes in two Lorenz-96 based experiments.

Apart from a few suggestions and questions below, the manuscript overall seems to be in a good shape to recommend it to **publish with a minor revision**.

> We thank the reviewer for his generous review.

1. P.1, L.15: **IES**

   The Ensemble Smoother (ES) van Leeuwen and Evensen (1996) is a non-sequential method. It involves no ensemble update during the whole run of the system. While ES produces an optimal analysis for linear systems only, it also seems to be a popular approach for nonlinear systems with stable dynamics, e.g. in reservoir modelling.

   In contrast, the EnRML is a sequential method specifically designed for nonlinear dynamics. I am therefore a bit lost when the authors say that EnRML is "also known as the iterative ensemble smoother (IES)", particularly considering that the paper in general is fairly thorough in reviewing the literature, and that Geir Evensen is one of the authors.

   If the above is correct, then using the term "ES-MDA" in regard to a sequential method also seems somewhat controversial.

   > The reviewer is probably thinking of Gu and Oliver [2007], which formulated EnRML sequentially. However, EnRML is usually employed for reservoir problems, where it is employed non-sequentially (or in "batch" mode). In this case, it is also sometimes known as the IES (especially among users of the ERT software).
   >
   > As mentioned in section 1.2, it is easy to formulate all "flavours" in sequential and batch formulations. This also includes ES-MDA. To try to resolve some confusion, we have moved the (single mention of the) IES acronym to section 1.2.

2. P1., L.18 and further: **Gu et al.**

   Should be "Gu and Oliver".

> Done, thanks.

**3. P.5, L.49: $\mathbf{C}_y$ may be identified as the prior covariance of observation**

Should be "as the prior covariance of innovation".

> We assume the "innovation" the reviewer is referring to is $\boldsymbol{d} := \boldsymbol{y} - \mathbf{M}\boldsymbol{\mu_x}$. Writing this as $\boldsymbol{d} = (\boldsymbol{y} - \mathbf{M}\boldsymbol{x}) + (\mathbf{M}\boldsymbol{x} - \mathbf{M}\boldsymbol{\mu_x})$, one may show that $\mathbb{C}\text{ov}(\boldsymbol{d}) = \mathbf{M}\mathbf{C_x}\mathbf{M}^\mathsf{T} + \mathbf{C_\delta}$, as the reviewer says. However, it is also clearly the prior covariance of $\boldsymbol{y} := \mathbf{M}\boldsymbol{x} + \boldsymbol{\delta}$. We have now included both naming options in the paper.

**4. P.5, L.61: Sakov et al. (2017)**

Should be "Sakov et al. (2018)".

> Done, thanks.

**5. P.7, L.26: Using the $\mathbf{C}_x$ in its stead is flawed and damaging because it is zero in the directions orthogonal to the ensemble subspace, so that its use would imply that the prior is assumed infinitely uncertain (i.e. flat) as opposed to infinitely certain (like a delta function) in those directions.**

I used to believe that using $\mathbf{C}_x^+$ is not a problem because the cost function is estimated in the ensemble subspace only.

> We're unsure if the reviewer is saying that he is not yet convinced by our arguments, or rather that he now *is* convinced.
>
> In the latter case: thanks!
>
> In the former case, consider that using $\boldsymbol{x}^\mathsf{T}\bar{\mathbf{C}}_x^+\boldsymbol{x}$ as the prior mismatch term would not penalize components of $\boldsymbol{x}$ outside of the ensemble subspace. Thus, these components of the posterior (ensemble) would just be set to those of the likelihood! This is clearly quite contrary to the correct posterior, which should not move from the prior (outside of the ensemble subspace). For simplicity, the above is based on assumptions that $\mathbf{C_\delta}$ and $\bar{\mathbf{C}}_x$ are proportional. The other cases are illustrated by Figure 1. In addition, we have conducted benchmarking investigations with Lorenz-96 which confirm that the Kalman-gain form is better.
>
> Note that the question does not even arise if the change of variables to $\boldsymbol{w}$ is done in the beginning of the derivation, because then the cost function is only formulated for the ensemble subspace. As noted in the manuscript, however, we chose to follow the "traditional" derivation precisely in order to highlight this and other issues.

**6. P.8, L.57: Bocquet et al.,2013**

Should be "Bocquet and Sakov, 2013".

> Done, thanks.

**7. P.14, L.12: where the data assimilation window has been fixed at L = 0.4, which is near optimal (cf. Bocquet et al., 2013, Figures 3 and 4)**

I am missing the point why $L = 0.4$ is near optimal.

**Figure 1:** Illustration of the consequence of using the pseudo-inverse to update the ensemble. Here, the state vector is two-dimensional, and the ensemble has two members. The correct update (red) stays in the ensemble subspace.

Consider Bocquet and Sakov [2013], Figure 3. We're investigating filtering performance, so their left pane is the relevant one. The method corresponding to our set-up is their `SDA IEnKS-N S=1`. It's optimal lag can be seen to lie between 15 and 20, but it's not much worse at 8. In model time, this correspond to $8 \times 0.05 = 0.4$, which is our setting. In their Figure 4, the optimal lag is 4, but the performance is not much worse for 2 (corresponding to our model time of 0.4).

Furthermore, we believe the relative/qualitative performance results are not very sensitive to this setting. We therefore decided to use only one, selecting a value that was fairly optimal, but slightly shorter (to be on the safe side, and because it is cheaper.),

8. Section 4.2

It would also be interesting to see results for a more nonlinear case $\Delta t_{obs} = 0.8$.

We have now included the case of $\Delta t_{\text{obs}} = 0.6$, which is a fairly extreme setting for data assimilation. It clearly demonstrates the need for more iterations when the nonlinearity is stronger. It also shows a wider disparity between the two types of smoothers: Gauss-Newton and MDA. This has been remarked on in the results text. We have also included the benchmark scores for the EnKF.

NB1: We have changed our definition of "analysis" RMSEs (before, it was defined by the application of the ultimate $\mathbf{W}_i$ to the forecast ensemble; now it is defined by the nonlinear propagation of the smoothed ensemble of iteration i). This has the effect of lowering the reported scores, although the algorithms themselves are unchanged.

NB2: we swapped the hollow/compact definition of the markers.

**RC2**

Major. This manuscript reformulates the algorithm of the original ensemble randomized maximum likelihood (EnRML), a stochastic EnKS, and compares its performance with other iterative schemes using the Lorenz-96 model. The manuscript is well written. I basically have no critical comment on the proposed method. However, I would like to recommend the authors performing some additional Lorenz-96 experiments to discuss differences between EnRML and iterative EnKS. For example, the following experiments would be interesting:

(1) experiments with imperfect observation error variance

(2) experiments with higher nonlinearily (e.g., larger obs error or infrequent assimilation)
(3) experiments with model imperfection

Those additional experiments, I believe, should improve the manuscript more.

> We thank the reviewer for his generous words and comments.
>
> - Concerning items 1 and 3: The distinction between EnRML/IEnKS (following our improvements to EnRML) is their stochastic/deterministic nature. Since both are derived from the same hidden Markov model (HMM) framework, without system error, there seems to be little reason to suspect that one would systematically be better suited to deal with it. We therefore prefer to keep the manuscript short and focused, and defer to other studies for the issue of system error.
>
> - Concerning item 2: Please see the response to P. Sakov's point 8, and M. Bocquet's point 26.

Minor. 1. Please add dimensions of matrices as Eqs. (13) and (14) so that we can follow equations easier.

2. I recommend the authors to add a schematic figure that simply illustrates the proposed method compared with the original EnRML and iterative EnKS.

3. "stochastic and deterministic ES-MDA" should be explained more.

> 1. Done, thanks.
>
> 2. We were not able to accomplish this to a satisfactory degree.
>
> 3. Done, thanks.

**RC3 (Marc Bocquet)**

This is a very nice paper. Taking inspiration from the deterministic IEnKS, it significantly clarifies the derivation of the EnRML which I personally always found a bit convoluted. In spite of all the technical details, it is a relief to see the simple and sleek algorithm in page 12 of the manuscript. It allows for an immediate comparison with the deterministic IEnKS and it makes sense even without going into the details of the derivation. Part (and only part) of the simplification is reminiscent of the simplification operated in Bocquet and Sakov (2012) onto the IEnKF algorithm of Sakov et al. (2012) (line 21 of the main algorithm in Appendix A) without fundamentally changing the method. This could be briefly mentioned. I have not checked the appendices which are very technical and would require too much time to confidently check.

> We thank the reviewer for his generous words. The "reminiscence" point is clarified in item 30.

1. The line numbering is inconsistent (unpractical at best)!

> Sorry! I don't know how it happened (. . . probably it's a compilation issue).
>
> PS: Also note that I'm not able to remove the pagewise option, which is part of copernicus.cls.

2. line 1: The parentheses around stochastic are unnecessary.

> I have gone back and forth on this a lot, actually. There is a subtle interplay of the phrasing (i) here, (ii) below, for the IEnKS, and (iii) in the title.

3. line 12: "reservoir" → "oil reservoir": I believe reservoir is not natural for the usual NPG reader.

> Done, thanks.

4. line 16: Readers unfamiliar with the EnRML may jump to the beginning of the derivation: section 2: But then, there would be no introduction at all for theses readers. They do deserve a few sentences! (the abstract does not count.)

> Done, thanks. Moreover, this resulted in the incorporation of the "layout" section as a paragraph here.

5. lines 27-28: Parentheses are missing around the equations' number.

> Done, thanks.

6. line 44: Bocquet and Sakov (2012) could be mentioned here, as it showed that the requirement for inverses in Sakov et al. (2012) was unnecessary.

> Done, thanks.

7. lines 50-52: I guess you are missing the key point which is that chaotic model require sequential assimilation, whereas oil reservoir models, although nonlinear (and non trivial), are not chaotic.

Done, thanks.

8. line 59: "some of the results" → "some of the mathematical results" (otherwise, the statement would seem odd).

Done, thanks.

9. line 64: This has also been discussed in Liu et al. (2017) and to some extent Morzfeld et al. (2018).

Done, thanks.

10. page 3, line 12: Please define ‖.

(Unless I'm missing something) this was/is defined right there (i.e. below eqn 3).

11. page 4, line 19: The section title is too generic and not consistent with the one chosen for section 2.3. "RML" would be more consistent.

Done, thanks.

12. page 4, line 30: "first-order" is ambiguous. Strictly speaking, this is second-order as a Taylor expansion but without second-order derivatives.

Done, thanks.

13. page 4, line 41: Please define $\mathbf{I}_M$.

Done, thanks.

14. Page 5, line 49: $\mathbf{C}_y$ is often called the innovation covariance matrix, or innovation statistics.

Please see our response to P. Sakov's point 3.

15. Page 5, line 50: "than the inversion to compute $\mathbf{C}$" → "than the inversion needed to compute $\mathbf{C}_\bullet$ in (7b)."

Done, thanks.

16. page 5, line 62: "yield" → "yields".

I like the subjunctive "flourish" in English, but ok.

17. page 5, line 68: "without the costly use of Metropolis-Hastings": More fundamentally this is due to the curse of dimensionality (see e.g., Liu et al. (2017)).

> Done, thanks.

18. page 7, line 6: "Maciejewski and Klein, 1985" → "as proven by ..." or use parentheses.

> Done, thanks.

19. page 7, Theorem 1: At the NPG level, this result can be seen and written down on the back of an envelope, and that is why nobody cares to do so. It is nice though that you show it rigorously but since the heuristic for this proof is adamant, this would not really be required in NPG. By that I mean that previous authors are not really guilty. Also, I believe that the heuristic derivation is not restricted to Gaussian distributions, is it?

> In order to hasten the exchange, we have discussed this item in a private correspondence with the reviewer. We thus established that the reviewer did not fully appreciate the result. Therefore, to better emphasize its qualities, the paragraph has been somewhat reorganized.

20. page 8, lines 41-48: This can be better understood thinking in terms of conditioning of the associated cost function.

> Done, thanks.

21. page 8, line 58: "the control vector" → "a control vector" right? there may be several.

> Yes, thanks.

22. page 9, line 90: "Inversely" → "Conversely".

> Done, thanks.

23. page 12, line 63: "pre-multiplying" is ambiguous. Say for instance on the right or on the left.

> Done, thanks (also did "post-multiplying")

24. page 13, line 85: There is also a quasi-static variant of the deterministic IEnKS that would be worth considering (Fillion et al., 2018). However the data assimilation window length your chose might not be long enough to do so.

> (Unless I'm missing something) this was/is mentioned in the associated footnote.

25. page 13, line 10: 10'000, do you mean 10,000?

> Yes, thanks.

26. **page 14, line 18: I would add the smoothing RMSE to the numerical study to muscle it up. Moreover, I noticed that it is often uneasy for colleagues to grasp that such filter yields better filtering RMSE. I would show both the filtering and smoothing RMSEs to tell that these methods do both with high accuracy.**

> We have included plots of the smoothing RMSE, and clarified the distinction to analysis RMSE further in the text.
>
> Also see the response to P. Sakov's point 8.

27. **page 14, line 31: You could mention that this is qualitatively if not quantitatively similar to the deterministic versus stochastic EnKF.**

> Done, thanks (it was/is also noted in the summary).

28. **Page 14, line 42: "has been subject of" → "has been the subject of".**

> Done, thanks.

29. **page 14, line 42: "tuning of the step length": what does it mean? I really have no clue.**

> Reformulated. Hopefully it makes sense now.

30. **page 15, line 48: This is really reminiscent of the improvement brought about by Bocquet and Sakov (2012) onto Sakov et al. (2012) in the context of the deterministic IEnKF/IEnKS.**

> As far as I can tell, avoiding the explicit computation of $\bar{\mathbf{M}}_i$ in favour of $\mathbf{Y}_i = \bar{\mathbf{M}}_i \mathbf{X}$, and its computation by an inverse transform, was already in place (in the deterministic/IEnKS context) in Sakov, Oliver, and Bertino [2012]. Thus, I believe you're referring to the product $(\mathbf{X}^{\mathsf{T}}\mathbf{X})^{+}\mathbf{X}^{\mathsf{T}}$ on line 21 of their algorithm, which was simplified by Bocquet and Sakov [2012] by the change of variables to $\boldsymbol{w}$ (I also faintly remember a presentation with an explicit reduction of the product, but I cannot find the source). It is accomplished in the same manner in our section 3.4, where the papers have now been appropriately cited.

31. **page 16, line 57: "do not go the length of" → "do not go to the length of".**

> Changed to "do not venture to"

32. **page 16, line 63: "prove" → "proves".**

> Done, thanks.

**References**

[revised manuscript text omitted]
\left(\mathcal{S}_n \cap \text{col}([\boldsymbol{\Upsilon}_{:n-1}, \ \mathbf{I}_{n:}])\right) < \dim(\mathcal{S}_n). \quad (40)$$

Since $\boldsymbol{v}_n$ is absolutely continuous with sampling space $\mathcal{S}_n$, equation (40) means that the probability that $\boldsymbol{v}_n \in \text{col}([\boldsymbol{\Upsilon}_{:n-1}, \ \mathbf{I}_{n:}])$ is zero. This implies $\text{H}_n$ a.s., establishing the induction. Identifying the final hypothesis ($\text{H}_N$) with $\text{rank}(\boldsymbol{\Upsilon}) = N$ concludes the proof. □

A corollary of Theorem 2 and Lemma 1 is that the ensemble subspace is also unchanged by the EnKF update. Note that both the prior ensemble and the model (involved through $\mathbf{Y}$) are arbitrary in Theorem 2. However, $\mathbf{C}_{\boldsymbol{\delta}}$ is assumed invertible. The result is therefore quite different from the topic discussed by Kepert [2004]; Evensen [2004], where rank deficiency arises due to a reduced-rank $\mathbf{C}_{\boldsymbol{\delta}}$.

**Conjecture 1.** *The rank of the ensemble is preserved by the EnRML update (a.s.) and $\mathbf{W}_i$ is invertible.*

We were not able to prove Conjecture 1, but it seems a logical extension of Theorem 2, and is supported by numerical trials. The following proofs utilize Conjecture 1, without which some projections will not vanish. Yet, even if Conjecture 1 should not hold (due to bugs, truncation, or really bad luck), Algorithm 1 is still valid and optimal, as discussed in sections 3.6.3 and 3.6.4.

**A.2  The transform matrix**

**Theorem 3.** $(\mathbf{X}^+\mathbf{X}_i)^+ = \mathbf{X}_i^+\mathbf{X}.$

*Proof.* Let $\mathbf{T} = \mathbf{X}^+\mathbf{X}_i$ and $\mathbf{S} = \mathbf{X}_i^+\mathbf{X}$. The following shows that $\mathbf{S}$ satisfies the four properties of the Moore-Penrose characterization of the pseudo-inverse of $\mathbf{T}$:

1. $\mathbf{TST} = (\mathbf{X}^+\mathbf{X}_i)(\mathbf{X}_i^+\mathbf{X})(\mathbf{X}^+\mathbf{X}_i)$
   $\quad = \mathbf{X}^+\boldsymbol{\Pi}_{\mathbf{X}_i}\boldsymbol{\Pi}_{\mathbf{X}}\mathbf{X}_i \qquad [\boldsymbol{\Pi}_{\mathbf{A}} = \mathbf{A}\mathbf{A}^+]$
   $\quad = \mathbf{X}^+\boldsymbol{\Pi}_{\mathbf{X}_i}\mathbf{X}_i \qquad\qquad [\text{Lemma } 1]$
   $\quad = \mathbf{T}. \qquad\qquad\qquad [\boldsymbol{\Pi}_{\mathbf{A}}\mathbf{A} = \mathbf{A}]$

2. $\mathbf{STS} = \mathbf{S}$, as may be shown similarly to point 1.

3. $\mathbf{TS} = \mathbf{X}^+\mathbf{X}$, as may be shown similarly to point 1, using Conjecture 1. The symmetry of $\mathbf{TS}$ follows from that of $\mathbf{X}^+\mathbf{X}$.

4. The symmetry of $\mathbf{ST}$ is shown as for point 3. □

This proof was heavily inspired by appendix A of Sakov et al. [2012]. However, our developments apply for EnRML (rather than the deterministic, square-root IEnKS). This means that $\mathbf{T}_i$ is not symmetric, which complicates the proof in that the focus must be on $\mathbf{X}^+\mathbf{X}_i$ rather than $\mathbf{X}_i^+$ alone. Our result also shows the equivalence of $\mathbf{S}^+$ and $\mathbf{T}$ in general, while the additional result of the vanishing projection matrix in the case of $N - 1 \leq M$ is treated separately, in appendix A.3.

**A.3  Proof of equation (34)**

**Lemma 2.** $\boldsymbol{\Omega}_i$ *is invertible (provided $\mathbf{W}_i$ is).*

*Proof.* We show that $\boldsymbol{\Omega}_i\boldsymbol{u} \neq 0$ for any $\boldsymbol{u} \neq 0$, where $\boldsymbol{\Omega}_i = \mathbf{W}_i\boldsymbol{\Pi}_{\mathbb{1}}^{\perp} + \boldsymbol{\Pi}_{\mathbb{1}}$. For $\boldsymbol{u} \in \text{col}(\mathbb{1})$: $\boldsymbol{\Omega}_i\boldsymbol{u} = \boldsymbol{u}$. For $\boldsymbol{u} \in \text{col}(\mathbb{1})^{\perp}$: $\boldsymbol{\Omega}_i\boldsymbol{u} = \mathbf{W}_i\boldsymbol{u} \neq 0$ (Conjecture 1). □

Recall that equation (33) was obtained by inserting $\mathbf{X}_i$ in the expression (30) for $\mathbf{T}_i$.  By contrast, the following inserts $\mathbf{X}$ from equation (35) in the expression (29) for $\mathbf{T}_i^+$

By equation (35) and Lemma 2, $\mathbf{X} = \mathbf{X}_i \mathbf{\Omega}_i^{-1}$ and so $\mathbf{T}_i^+ = \mathbf{\Pi}_{\mathbf{X}_i} \mathbf{\Omega}_i^{-1}$. We now re-introduce $\mathbf{\
[revised manuscript text omitted]

Scott Kirkpatrick, C. Daniel Gelatt, and Mario P. Vecchi. Optimization by simulated annealing. *science*, 220 (4598):671–680, 1983.

Peter K. Kitanidis. Quasi-linear geostatistical theory for inversing. *Water resources research*, 31(10):2411–2419, 1995.

Duc H. Le, Alexandre A. Emerick, and Albert C. Reynolds. An adaptive ensemble smoother with multiple data assimilation for assisted history matching. *SPE Journal*, 21(06):2–195, 2016.

Jun S. Liu. Siegel's formula via Stein's identities. *Statistics & Probability Letters*, 21(3):247–251, 1994.

Y. Liu, J.-M. Haussaire, M. Bocquet, Y. Roustan, O. Saunier, and A. Mathieu. Uncertainty quantification of pollutant source retrieval: comparison of bayesian methods with application to the chernobyl and fukushima daiichi accidental releases of radionuclides. *Quarterly Journal of the Royal Meteorological Society*, 143(708):2886–2901, 2017.

David M. Livings, Sarah L. Dance, and Nancy K. Nichols. Unbiased ensemble square root filters. *Physica D: Nonlinear Phenomena*, 237(8):1021–1028, 2008.

Andrew C. Lorenc. Development of an operational variational assimilation scheme. *Journal of the Meteorological Society of Japan. Series. II*, 75 (Special issue: data assimilation in meteorology and oceanography: theory and practice)(1B):339–346, 1997.

Edward N. Lorenz. Predictability: A problem partly solved. In *Proc. ECMWF Seminar on Predictability*, volume 1, pages 1–18, Reading, UK, 1996.

Xiaodong Luo, Andreas S. Stordal, Rolf J. Lorentzen, and Geir Naevdal. Iterative ensemble smoother as an approximate solution to a regularized minimum-average-cost problem: Theory and applications. *SPE Journal*, 20(05):962–982, 2015.

Xiang Ma, Gill Hetz, Xiaochen Wang, Linfeng Bi, Dave Stern, and Nazish Hoda. A robust iterative ensemble smoother method for efficient history matching and uncertainty quantification. In *SPE Reservoir Simulation Conference*. Society of Petroleum Engineers, 2017.

Anthony A. Maciejewski and Charles A. Klein. Obstacle avoidance for kinematically redundant manipulators in dynamically varying environments. *The international journal of robotics research*, 4(3):109–117, 1985.

J. Mandel, E. Bergou, S. Gürol, S. Gratton, and I. Kasanický. Hybrid Levenberg-Marquardt and weak-constraint ensemble Kalman smoother method. *Nonlinear Processes in Geophysics*, 23(2):59–73, 2016.

M. Morzfeld, D. Hodyss, and J. Poterjoy. Variational particle smoothers and their localization. *Quarterly Journal of the Royal Meteorological Society*, 144(712): 806–825, 2018.

Robb J. Muirhead. *Aspects of multivariate statistical theory.* John Wiley & Sons, Inc., New York, 1982. Wiley Series in Probability and Mathematical Statistics.

Dean S. Oliver. On conditional simulation to inaccurate data. *Mathematical Geology*, 28(6):811–817, 1996.

Dean S. Oliver. Metropolized randomized maximum likelihood for improved sampling from multimodal distributions. *SIAM/ASA Journal on Uncertainty Quantification*, 5(1):259–277, 2017.

Dean S. Oliver and Yan Chen. Recent progress on reservoir history matching: a review. *Computational Geosciences*, 15(1):185–221, 2011.

Dean S. Oliver, Albert C. Reynolds, and Ning Liu. *Inverse Theory for Petroleum Reservoir Characterization and History Matching.* Cambridge University Press, 2008.

Edward Ott, Brian R. Hunt, Istvan Szunyogh, Aleksey V. Zimin, Eric J. Kostelich, Matteo Corazza, Eugenia Kalnay, D. J. Patil, and James A. Yorke. A local ensemble Kalman filter for atmospheric data assimilation. *Tellus A*, 56(5):415–428, 2004.

Carlos Pires, Robert Vautard, and Olivier Talagrand. On extending the limits of variational assimilation in nonlinear chaotic systems. *Tellus A: Dynamic Meteorology and Oceanography*, 48(1):96–121, 1996.

Patrick N. Raanes, Marc Bocquet, and Alberto Carrassi. Adaptive covariance inflation in the ensemble Kalman filter by Gaussian scale mixtures. *Quarterly Journal of the Royal Meteorological Society*, 145(718):53–75, 2019. doi: 10.1002/qj.3386.

Javad Rafiee and Albert C. Reynolds. Theoretical and efficient practical procedures for the generation of inflation factors for ES-MDA. *Inverse Problems*, 33(11): 115003, 2017.

A. C. Reynolds, M. Zafari, and G. Li. Iterative forms of the ensemble Kalman filter. In *10th European Conference on the Mathematics of Oil Recovery*, 2006.

William Sacher and Peter Bartello. Sampling errors in ensemble Kalman filtering. Part I: Theory. *Monthly Weather Review*, 136(8):3035–3049, 2008.

Pavel Sakov and Laurent Bertino. Relation between two common localisation methods for the EnKF. *Computational Geosciences*, 15(2):225–237, 2011.

Pavel Sakov and Peter R. Oke. Implications of the form of the ensemble transformation in the ensemble square root filters. *Monthly Weather Review*, 136(3):1042–1053, 2008.

Pavel Sakov, Dean S. Oliver, and Laurent Bertino. An iterative EnKF for strongly nonlinear systems. *Monthly Weather Review*, 140(6):1988–2004, 2012.

Pavel Sakov, Jean-Matthieu Haussaire, and Marc Bocquet. An iterative ensemble Kalman filter in the presence of additive model error. *Quarterly Journal of the Royal Meteorological Society*, 144(713):1297–1309, 2018.

Andreas S. Stordal. Iterative Bayesian inversion with Gaussian mixtures: finite sample implementation and large sample asymptotics. *Computational Geosciences*, 19(1):1–15, 2015.

Xiangjun Tian, Zhenghui Xie, and Aiguo Dai. An ensemble-based explicit four-dimensional variational assimilation method. *Journal of Geophysical Research: Atmospheres*, 113(D21), 2008.

Lloyd N. Trefethen and David Bau, III. *Numerical linear algebra.* Society for Industrial and Applied Mathematics (SIAM), Philadelphia, PA, 1997.

Peter Jan van Leeuwen. Comment on "Data assimilation using an ensemble Kalman filter technique". *Monthly Weather Review*, 127(6):1374–1377, 1999.

Mohammad Zafari and Albert Coburn Reynolds. Assessing the uncertainty in reservoir description and performance predictions with the ensemble Kalman filter. Master's thesis, University of Tulsa, 2005.

Milija Zupanski. Maximum likelihood ensemble filter: Theoretical aspects. *Monthly Weather Review*, 133(6): 1710–1726, 2005.